# zfh2 controls progenitor cell activation and differentiation in the adult *Drosophila* intestinal absorptive lineage

**Sebastian E. Rojas Villa[1], Fanju W. Meng[2], Benoît Biteau[2]***

**1** Department of Biology, University of Rochester, Rochester, New York, United States of America,
**2** Department of Biomedical Genetics, University of Rochester Medical Center, Rochester, New York, United States of America

\* benoit_biteau@urmc.rochester.edu

**Data Availability Statement:** All relevant data are within the manuscript and its Supporting Information files.

**Funding:** Work in the Biteau laboratory is supported by the National Institutes of Health

## Abstract

Many tissues rely on resident stem cell population to maintain homeostasis. The balance between cell proliferation and differentiation is critical to permit tissue regeneration and prevent dysplasia, particularly following tissue damage. Thus, understanding the cellular processes and genetic programs that coordinate these processes is essential. Here, we report that the conserved transcription factor zfh2 is specifically expressed in *Drosophila* adult intestinal stem cell and progenitors and is a critical regulator of cell differentiation in this lineage. We show that zfh2 expression is required and sufficient to drive the activation of enteroblasts, the non-proliferative progenitors of absorptive cells. This transition is characterized by the transient formation of thin membrane protrusions, morphological changes characteristic of migratory cells and compensatory stem cell proliferation. We found that zfh2 acts in parallel to insulin signaling and upstream of the TOR growth-promoting pathway during early differentiation. Finally, maintaining zfh2 expression in late enteroblasts blocks terminal differentiation and leads to the formation of highly dysplastic lesions, defining a new late cell differentiation transition. Together, our study greatly improves our understanding of the cascade of cellular changes and regulatory steps that control differentiation in the adult fly midgut and identifies zfh2 as a major player in these processes.

## Author summary

The ability of stem cells to produce functional cells, through the process of differentiation, is critical to maintain the integrity and function of many adult organs. Therefore, describing the molecular and cellular mechanisms that control cell differentiation is an essential part in understanding tissue regeneration, as well as diseases such as cancer or degenerative syndromes. For over a decade, the intestine of the fruitfly *Drosophila* has served as a model to study adult tissue stem cells in a genetically amenable organism. Here we report a novel function for the conserved transcription factor zfh2, ATBF1 in mammals, and demonstrate that it controls an essential cell fate transition during early differentiation in the fly intestine. We also show that abnormal expression of this regulator leads to the

(5R01GM108712-05 to B.B) and the Ellison Medical Foundation (AG-NS-0990-13 to B.B.). The funders had no role in study design, data collection and analysis, decision to publish, or preparation of the manuscript.

**Competing interests:** The authors have declared that no competing interests exist.

rapid formation of aggressive tumors. Our work sheds new light on the function of zfh2 and related factors in the control of cell identity and will likely help us and others formulate new hypotheses regarding the role of these transcription factors in cancer.

## Introduction

In adult somatic tissues, most cells are differentiated and have specialized roles in maintaining proper tissue and organismal function. During normal turnover or in response to damage, resident adult somatic stem cells can undergo cell division and differentiation, in order to maintain tissue integrity and the proper number of differentiated cells [1]. Characterizing the conserved signaling pathways controlling somatic stem cell activity has been one of the main focus of developmental biology and regenerative medicine. However, studying the process of differentiation *in vivo* remains difficult, as it requires characterizing the successive phenotypical changes associated with cell fate transitions and how signaling pathways and transcriptional networks controls these coordinated cellular changes.

The *Drosophila melanogaster* adult intestine offers a powerful and tractable model to study adult somatic cell differentiation *in vivo*. The intestinal stem cell (ISC) can give rise to the full lineage of cells present in this epithelial tissue. ISC can undergo non-symmetrical cell division and give rise to the progenitors of enteroendocrine cell (EEs) and enterocytes (ECs), pre-EEs and enteroblasts (EBs) respectively. The EB then undergoes growth and terminal differentiation into the large absorptive cell, the enterocyte [2–4]. In the last decade studies have mostly focused on characterizing the mechanisms that control stem cell proliferation in this tissue. The adult *Drosophila* intestinal epithelium is a very quiescent tissue, where ISC proliferation occurs rarely under homeostasis but is rapidly induced upon tissue damage or infection, via a host of highly conserved signaling pathways [5–8]. Less is known about signaling pathways controlling ISC and EB differentiation. Studies have shown that the Target of Rapamycin (Tor) signaling pathway is sufficient to induce intestinal progenitor growth and required for differentiation in the EC fate [9, 10]. Similarly, Insulin signaling acts cell autonomously in EB regulating its growth and differentiation [11]. Finally, another study has recently suggested that, under stress conditions, EC cell fate is regulated by EGFR signaling in a Tor-independent manner [12].

In parallel to the genetic analysis of the regulator of EB differentiation, recent work has also focused on characterizing the phenotypical transitions associated with EB differentiation and the effect differentiating EB on tissue homeostasis. Antonello et al. showed that EBs can remain undifferentiated, in a dormant-like state, for long periods of time, and that cell death and tissue damage induces EB activation and eventually terminal differentiation into ECs. The activation of dormant EBs is not only coupled with cell growth but also with changes in cell morphology, as EBs gain structures that are actin rich lamellipodia-like, and with formation of thin membrane protrusions. These structures have been hypothesized to mediate activated EB migration to the site of tissue damage before terminal differentiation into ECs [13]. Other studies have highlighted the effect that dormant versus activated EBs have on ISC proliferation. Small dormant EBs inhibit neighboring ISC proliferation via cell/cell contact inhibition, mediated by high levels of E-Cadherin [11], while blocking terminal differentiation of EBs into ECs leads to secretion of mitogens inducing ISC proliferation [14, 15]. Despite the progress in our understanding of adult *Drosophila* intestinal progenitor differentiation very little is known of the sequence of events and mechanisms associated with EB activation.

Our work characterizes zfh2, a conserved zinc finger homeodomain transcription factor [16] as a key player in the maintenance of adult intestinal epithelial homeostasis. Zfh2 has previously been studied in the context of *Drosophila* development: during wing development nubbin/Pdm1 represses zfh2 expression, moreover zfh2 expression in this tissue allows Wg activation. Impairing zfh2 expression in this tissue leads to large deletions in the wing hinge and general disorganization of the wing veins [17, 18]. In the context of tarsal segment development zfh2 expression is controlled by notch signaling and is required for proper leg size and to form the leg joints between the fourth and the fifth tarsal segment [19]. zfh2 is also expressed in a subset of serotonergic and dopaminergic neurons during larval stage, but its function remains unknown [20]. To date, very little is known about the function of zfh2 in adult tissues, but a recent paper identified a fat body specific role of zfh2 in hypercapnic immune regulation [21]. Interestingly, zfh2 is highly conserved, its mammalian homolog ATBF1 has been proposed as a possible tumor suppressor gene [22]. Altered levels of ATBF1 expression have been correlated with invasiveness and high tumorigenesis in prostate, breast and gastric cancer [23–25], but the exact mechanism by which ATBF1 may control these processes is not well understood. Another transcription factor has been recently described as another zfh2 homolog: ZFHX4. Mutations in ZFHX4 have been associated with esophageal squamous cell carcinoma migration and invasiveness, but the mechanisms remain unknown [26, 27].

In this work we identify zfh2 as a critical player in EB activation and differentiation. We found that zfh2 is expressed in intestinal progenitors, required for EB activation and its overexpression is sufficient to induce cellular phenotypes associated with this process. Our results also establish a hierarchy of genetic requirements and phenotypes associated with EB activation and suggest that the EB needs to be primed by zfh2-dependent mechanisms during activation prior to its growth and differentiation. Finally, we show that zfh2 expression needs to be turned off to allow EB terminal differentiation into ECs. Altogether, our work greatly improves our understanding of the process of EB activation and the role that zfh2 plays in this cellular transition.

## Results

### zfh2 is expressed in stem cells and enteroblasts in the adult *Drosophila* intestinal epithelium

zfh2 expression in the intestinal epithelium and intestinal progenitors has been previously detected by large-scale transcriptomic analyses [28, 29]. Before investigating the possible role of zfh2 in the maintenance of intestinal homeostasis, we first confirmed that this protein is expressed in progenitors in the adult gut. Using immunohistochemistry and an antibody directed against zfh2, we detected the zfh2 protein in all escargot-positive (esg+) cells, ISCs and EBs (Fig 1A). To confirm the specificity of this staining, we used the temperature sensitive driver esgGal4,UAS-GFP;tubGal80ts (esgGal4ts>GFP) to express two independent dsRNA constructs against zfh2 and observed that the signal is completely lost (Fig 1A). When using specific markers to distinguish between ISCs (Sox21a-positive, Su(H)GBEGal4>GFP-negative) and EBs (Sox21a-positive, Su(H)GBEGal4>GFP-positive), we found no difference in zfh2 expression levels between these two cell types under normal conditions or in response to stress (S1A, S1B and S1C Fig). Because Notch signaling is inactive in ISCs and active in EBs [30], this suggests that, unlike during leg development [19], in the intestinal lineage, zfh2 expression is not significantly affected by Notch activity.

To further confirm that zfh2 is exclusively transcribed in ISCs and EBS in the adult intestine, we used two independent transcriptional reporters: Zfh2Gal4^GMR73G11 (a transgenic construct in which Gal4 is under the control of a 1kb enhancer from the zfh2 locus) and

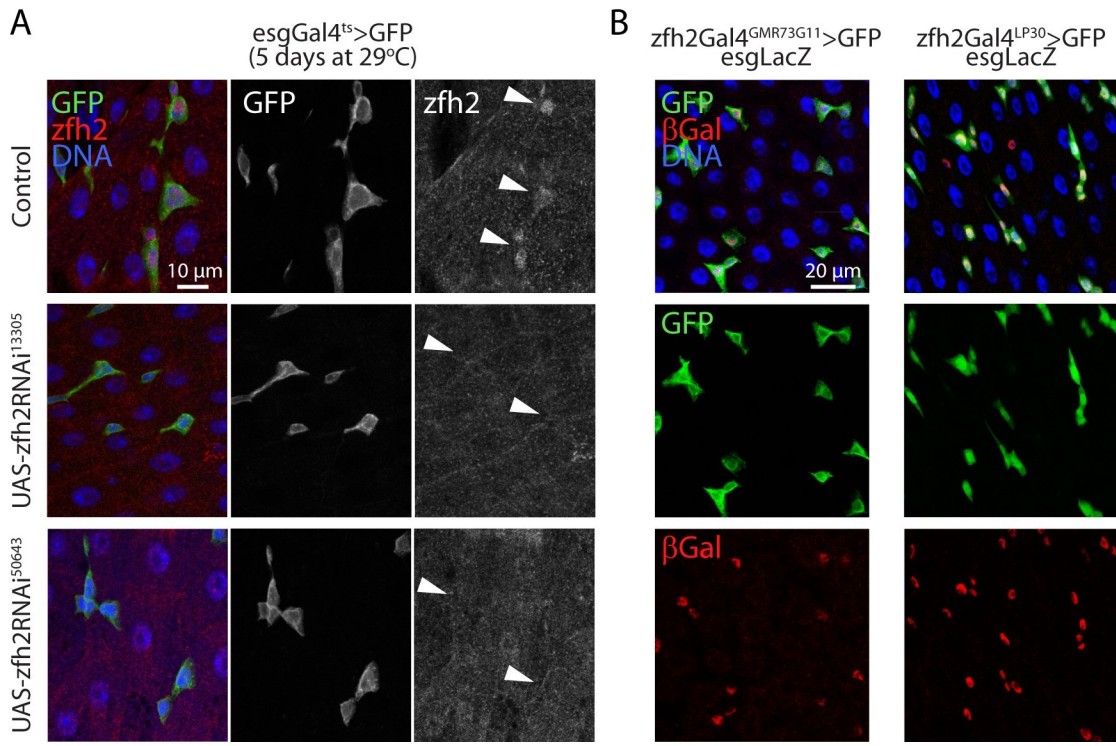

**Fig 1. zfh2 is expressed in adult intestinal progenitors.** (A) Representative confocal images of the posterior midgut. ISCs and EBs are labeled via esgGal4ts>GFP. zfh2 protein is detected via immunohistochemistry only in GFP-positive cells (indicated by the arrowhead). zfh2 protein is not detected when 2 independent dsRNAs directed against zfh2 are expressed using the esgGal4ts driver. (B) Representative confocal images of the posterior midgut of two zfh2 transcriptional reporters driving GFP expression (zfh2Gal4$^{GMR73G11}$ and Zfh2Gal4$^{LP30}$). βGalactosidase, from the reporter line esgLacZ identifies ISCs and EBs.

Zfh2Gal4$^{LP30}$ (a Gal4-containing transposon insertion in the zfh2 locus) [31]. We combined these reporters with the esg-LacZ reporter and used immunohistochemistry to label β-galacto-sidase expressing cells. Both transcriptional reporters are active specifically in all esg+ cells, further confirming that zfh2 is transcriptionally active only in ISCs and EBs in the adult mid-gut epithelium (Fig 1B).

## Manipulating zfh2 expression levels specifically affects enteroblast size

To investigate the role of zfh2 in the adult intestinal epithelium, we first used temperature sen-sitive driver esgGal4ts to knock down zfh2 in both ISCs and EBs, using multiple independent dsRNAi constructs. In these conditions, we observed no significant change in the overall com-position of the epithelium (i.e. proportion of ISCs, EBs, ECs or EEs) (S2A Fig). However, we found that manipulating zfh2 strongly affects EB cell size. Indeed, we measured cell area of esg +Dl+ ISCs and esg+Dl- EBs and found that knocking-down zfh2 for 3 days weakly affects ISC size but significantly reduces the average size of EBs (Fig 2A and 2B). Conversely, over-expressing zfh2, using the UAS-zfh2$^{EAB}$-mCherry insertion line [18] and the esgGal4ts driver, does not affect ISC size but significantly increases EB size (Fig 2A and 2B). To test if this effect is cell-autonomous, we repeated the same manipulations with the EB-specific driver Su(H) GBEGal4,tubGal80ts (GBEGal4ts). We found that knocking-down zfh2 in this manner leads to a decrease in the size of EBs, while over-expressing zfh2 leads to an increase in cell size (Fig 2C and 2D). Finally, in order to consider possible shape changes induced by manipulating

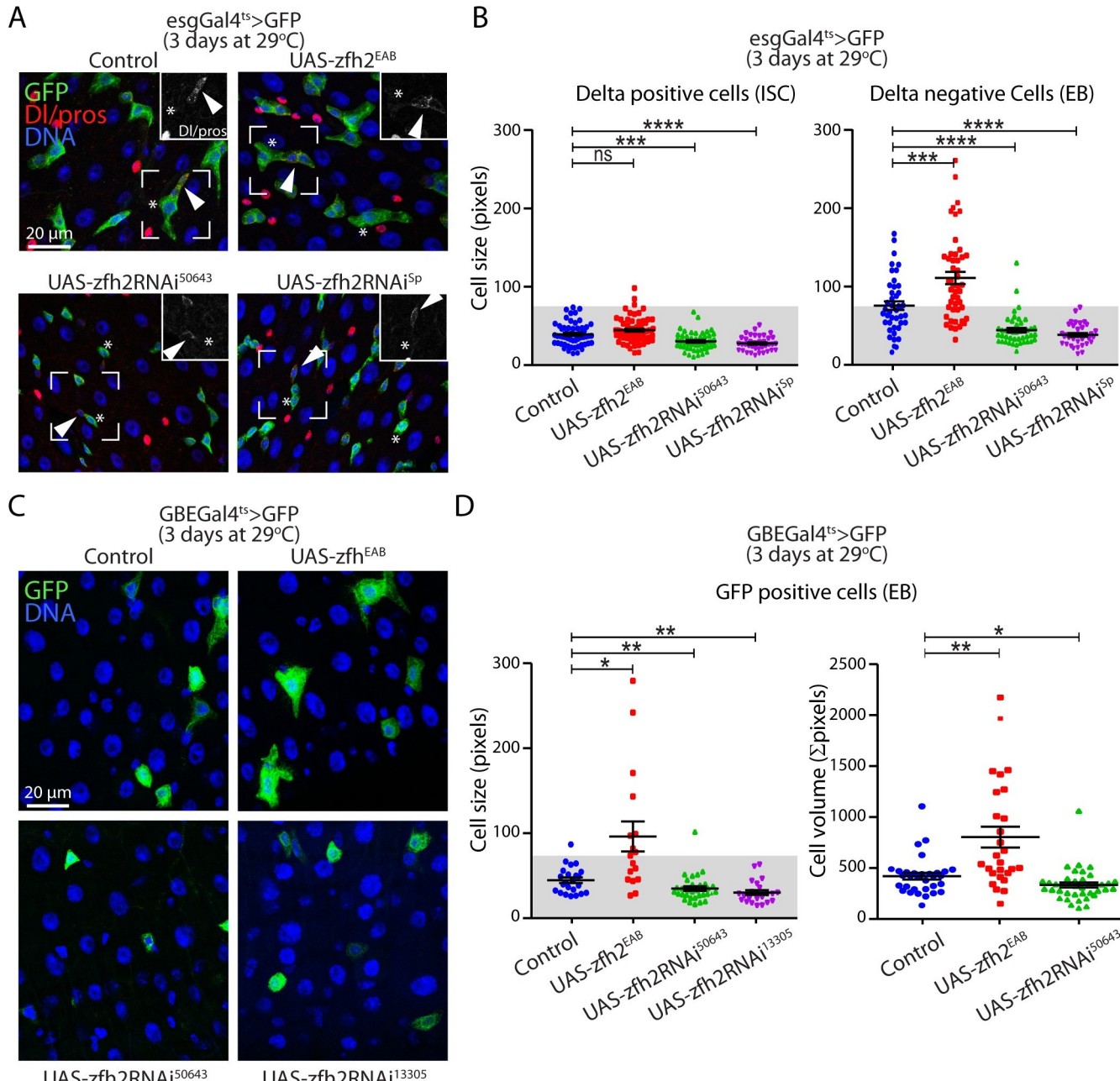

**Fig 2. zfh2 controls enteroblast growth.** (A) Representative confocal images of the posterior midgut. ISCs and EBs express GFP (esgGal4ts>GFP). ISCs and enteroendocrine cells are labeled via immunohistochemistry against Delta (red, membrane) and prospero (red, nuclear) respectively. ISCs are identified as GFP-positive, Delta-positive cells (arrowheads) and EBs are identified as GFP-positive, delta-negative cells (asterisks). Inserts show the isolated Delta/Prospero channel of the boxed regions to highlight the difference in Delta staining between ISCs and EBs. zfh2 over-expression, by driving the UAS-zfh2EAB transgene using esgGal4ts, is sufficient to increase EB cell size. zfh2 knock-down, by driving dsRNAs against zfh2, leads to smaller EB size. (B) Cell size of ISCs and EBs is quantified by measuring cell area of individual cells. (C) EBs are specifically labeled by GBEGal4ts>mCD8GFP expression. zfh2 is over-expressed by driving the UAS-zfh2EAB transgene using GBEGal4ts. zfh2 is knocked-down by driving dsRNA against zfh2 using GBEGal4ts. (D) Cell size of EBs is quantified by measuring cell area of individual cells. Cell volume of EBs is quantified by adding cell areas through entire z-stacks containing EBs. In B and D values are presented as average +/- s.e.m. and p-values are calculated using a two-tailed Student's t-test.

zfh2 that may confound our analysis of cell size based solely on cell area, we directly estimated cell volume by adding the cell area through confocal imaging stacks encompassing entire EBs.

Consistent with our previous observations, zfh2 knock-down results in smaller EBs, while zfh2 over-expression significantly increases EB size (Fig 2D).

During tissue turnover or regeneration, EB growth is part of the EB differentiation process. Newly formed EBs can remain in a dormant state for an extended period before undergoing growth and endoreplication as part of a process known as EB activation, eventually leading to terminal differentiation into ECs [13]. Dormant EBs are similar in size and shape to its diploid neighboring ISCs. Comparing ISCs and EBs size, we observed that around 50% of wild-type EBs have a cell area comparable to small diploid ISCs, consistent with the high percentage of dormant EBs. However, we found that knocking down zfh2 in EBs increases the number of small EBs (93% and 97% of EBs with sizes similar to wild type ISCs) and, conversely, over-expressing zfh2 leads to an increase in the number of EBs larger than ISCs (from 47% in the control to 73% after zfh2 over-expression) (Fig 2A and 2B). During the process of EB activation and differentiation, EB growth is coupled with endoreplication. In the same zfh2 knock-down conditions, we observed an accumulation of EBs with small nuclei, identical in size to diploid ISCs, supporting the notion that zfh2 is required for EB endoreplication (S2B and S2C Fig).

Altogether these data strongly suggest that zfh2 is required and sufficient for EB activation.

## Characterization of the enteroblast activation and the role of zfh2

The morphological and transcriptional changes associated with EB activation have recently started to be described [13]. However, no rigorous method to quantify the activation state of EBs have been established. Therefore, to investigate the role of zfh2 specifically in the activation process, we first established ways to measure and quantify the activation state of EBs. Dormant EBs have been described as similar to ISC in their oval-like morphology. Upon activation, EBs take a more elongated and irregular shape, with the formation of pseudopodia and thin membrane protrusions that can only be observed in non-fixed tissue [13]. We first used DSS (Dextran Sulfate Sodium), a chemical stressor that acts by disorganizing the basement membrane [5], to induce synchronous and rapid activation of the majority of EBs in the epithelium. When we imaged EBs expressing membrane-bound GFP without fixation, we observed an increase in the presence of EBs with irregular shapes and thin membrane protrusions 6 hours after DSS treatment (Fig 3A). These protrusions are not present after a 12hour treatment, suggesting that these morphological changes are transient and associated with early EB activation (Fig 3A). To quantitatively characterize the activation state of EBs, we measured cell circularity and categorized EBs based on the presence of membrane protrusions, and confirmed that EBs are activated (low circularity and high number of protrusions) in the intestine of flies fed DSS for 6 hours (Fig 3B and 3C). Next, we asked whether similar EB activation can be detected in response to other stimuli and found that the same EB morphological changes can be detected 3 to 4 hours after infection with the bacterium Ecc15 or exposure to oxidative stressor paraquat (S3A, S3B and S3C Fig). Finally, we confirmed that the detected membrane protrusions are actin-rich, as previously reported [13], using the moesin-GFP reporter (S3D Fig).

Using this methodology, we next assessed the effect of manipulating zfh2 expression in EB activation. Using the GBEGal4ts driver to express two zfh2RNAi construct, we found that knocking-down zfh2 prevents DSS-induced EB activation as shown by the high circularity and absence of protrusion in GBEGal4ts>zfh2RNAi EBs (Fig 3D, 3E and 3F). Conversely, over-expressing zfh2 under normal culture conditions is sufficient to decrease EB circularity and induced the formation of cellular protrusions (Fig 3G, 3H and 3I). Besides changes in morphology, activated EBs have been shown to express high levels of the transcription factor

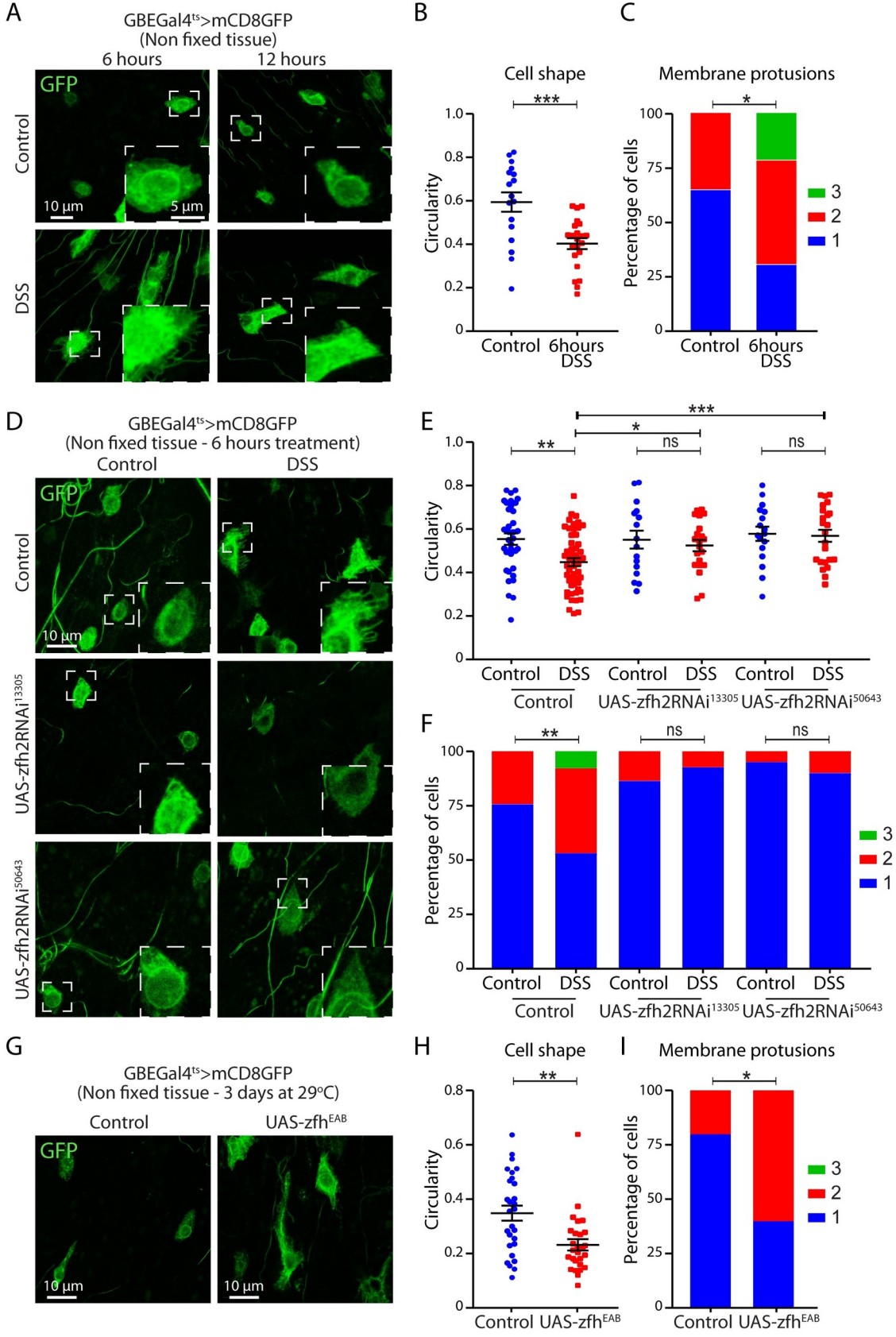

**Fig 3. zfh2 regulates enteroblast activation.** (A, D and G) Representative confocal images of non-fixed posterior midgut epithelia. EBs are labeled by GBEGal4ts>mCD8GFP. Note that autofluorescence signal from trachea is also visible in some images. (A) Stress mediated EB activation is induced by DSS exposure for either 6 or 12h. Morphological and size changes are apparent in EBs 6 hours after treatment. (B) Changes in cell morphology 6 hours after stress induction are quantified by measuring circularity of individual cells. (C) Presence or absence of thin membrane protrusions are quantified in individual cells 6h after DSS mediated stress. Categories are defined as: Type 1 cells have none to few membrane protrusions (<10), type 2 cells have many (>10) thin membrane protrusions, type 3 cells have many long and thick membrane protrusions (>2.5 μm in length). 6 hours of DSS treatment are sufficient to induce an increase in the percentage of cells with membrane protrusions. (D) zfh2 is knocked-down by driving dsRNAs against zfh2 using GBEGal4ts. zfh2 knock-down is sufficient to block DSS-induced changes in cell morphology (E) and the increase in cells with thin membrane protrusions (F). (G) zfh2 is over-expressed by driving the UAS-zfh2EAB transgene using GBEGal4ts. zfh2 over-expression is sufficient to decrease circularity (H) and increase the number of EBs with membrane protrusions (I). In B, E and H values are presented as average +/- s.e.m, and p-values are calculated using a two-tailed Student's t-test. In C, F an I p-values are calculated using the Mann-Whitney test.

Sox21a [14, 15, 32]. To confirm that zfh2 regulates activation, we investigated the expression of Sox21a in zfh2 gain-of-function and found that over-expressing zfh2 is sufficient to induce Sox21a expression in EBs (S3E and S3F Fig).

Together, our results demonstrate that zfh2 expression in EBs is required for their activation and that increased zfh2 expression is sufficient promote EB activation.

## Influence of zfh2-mediated enteroblast activation on tissue homeostasis

EB activation is required prior to EB differentiation into ECs. Therefore, we predicted that zfh2 is also required for terminal differentiation. To test this notion, we used the G-TRACE expression system [33] to permanently label and knock-down zfh2 in EB lineages and immunostaining against the transcription factor Pdm1 as a marker of terminal differentiation. In absence of stress, we identified a limited number of newly differentiated ECs within a 6 days period after labeling induction (Fig 4A). However, after DSS treatment in control animals, we detected an increase in newly differentiated Pdm1+ ECs, allowing us to measure a significant differentiation index (number of ECs normalized by the number of EBs). When zfh2RNAi is expressed in EBs, no significant differentiation was measured after stress exposure, confirming that zfh2-dependent EB activation is required prior to differentiation (Fig 4A).

In addition to differentiation, the activation state of EBs also affects ISC proliferation in a non-cell autonomous manner. Small dormant EBs inhibit neighboring ISC proliferation via cell/cell contact interaction mediated by high levels of the adhesion protein, E-Cadherin [11]. Conversely, blocking EB terminal differentiation into EC, and thus likely arresting EBs in an activated state, induces ISC proliferation via the JAK/STAT and EGFR signaling [15, 32]. Consistent with the notion that knocking-down zfh2 arrests EBs in a dormant state, we observed that expressing zfh2RNAi in an EB-specific manner is sufficient to partially impair ISC proliferation induced by DSS (Fig 4B). On the contrary, promoting EB activation by over-expression of zfh2 specifically in EBs (GBEGal4ts driver) induces ISC proliferation, while over-expression in ISCs (DlGal4ts and ISCGal4ts) has no effect (Fig 4C).

These analyses of differentiation and proliferation further demonstrate that zfh2 significantly impact tissue homeostasis in the intestinal epithelium by regulating EB activation.

## zfh2-mediated activation is required for cell growth in enteroblasts

We found that zfh2 controls EB activation and growth. Thus, we asked whether promoting growth is sufficient to induce activation or rather whether zfh2-dependent EB activation is required for cell growth. Several signaling pathways have been implicated in EB growth and terminal differentiation, including Tor, insulin and Ras signaling [9–12]. The function of the Tor pathway in EB growth is best characterized. Thus, we first asked whether zfh2 in EBs can

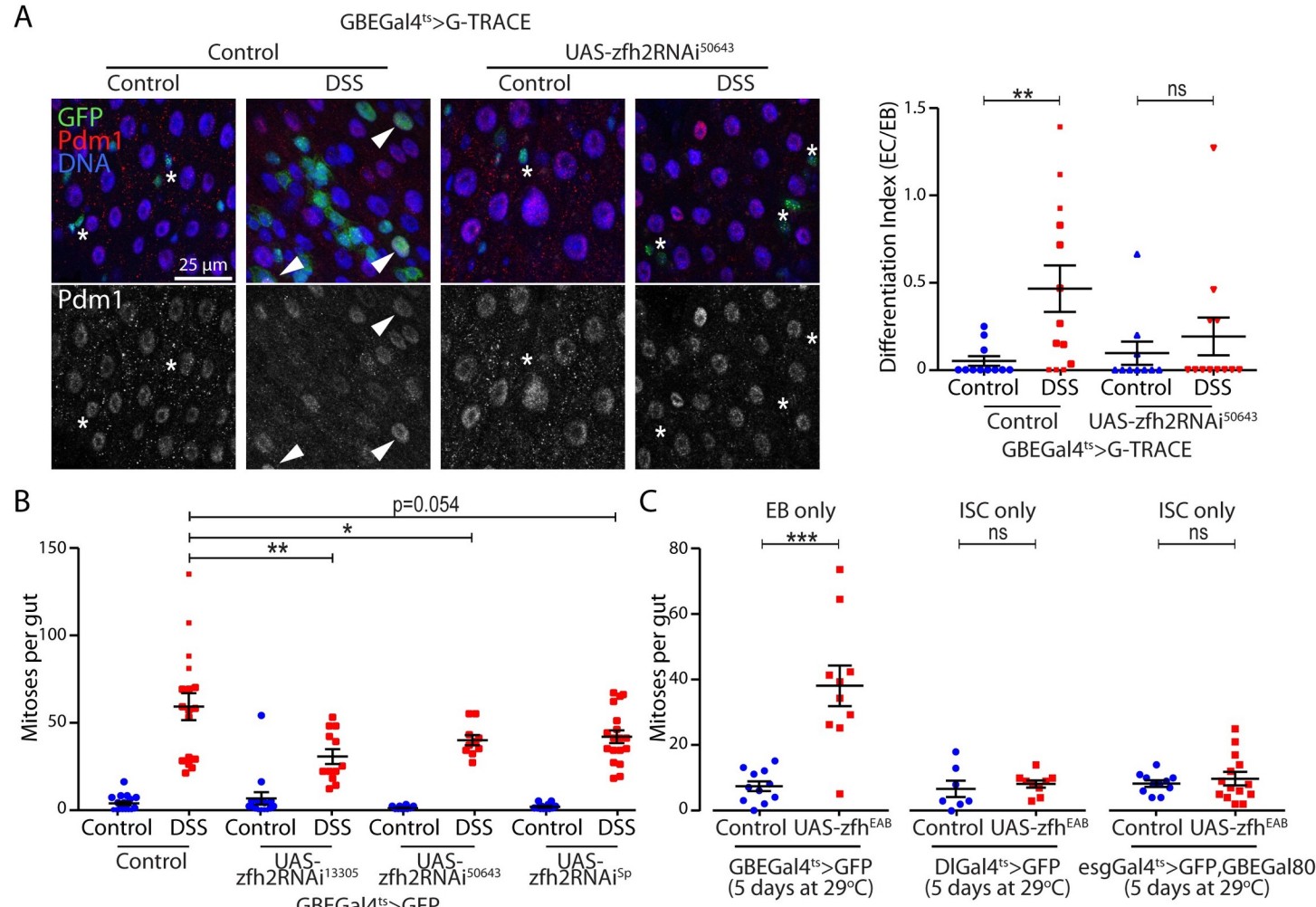

**Fig 4. zfh2 is essential to maintain intestinal homeostasis.** (A) The G-TRACE lineage tracing reporter is driven by the GBEGal4ts to permanently label EBs with GFP. ECs are labeled by immunohistochemistry against Pdm1. Flies are treated with DSS for 24 hours to induce EB activation and then left to recover under normal conditions for 48 hours. EBs are identified as small GFP-positive Pdm1-negative cells (asterisk), while newly differentiated EC are large GFP-positive Pdm1-positive cells (arrowhead). Differentiation ratio is measured as the ratio between newly differentiated EC (GFP+ PDM1+ cells) and EB (GFP+ PDM1- cells) in the posterior midgut. Each value represents the average of 2 ROI per midgut. Knocking-down zfh2 by driving dsRNA blocks terminal differentiation of EB into EC. (B) zfh2 is knocked-down in EBs by driving dsRNA using GBEGal4ts. Flies were fed either DSS or Sucrose for 48 hours before dissection and fixation. Mitoses per gut are measured using immunohistochemistry against phospho-Histone H3. zfh2 knock-down in EBs impairs DSS-mediated ISC proliferation non-cell autonomously. (C) zfh2 is over-expressed in EBs by driving the UAS-zfh2EAB transgene, using GBEGal4ts, or in ISCs using the DlGal4ts or esgGal4ts,GBEGal80 drivers. zfh2 is sufficient to promote ISC proliferation non-cell autonomously. In A, B and C values are presented as average +/- s.e.m, and p-values are calculated using a two-tailed Student's t-test.

induce Tor activity. To this end, we measured Tor activity by quantifying protein levels of the downstream target of the Tor kinase complex 1 (Torc1), phospho-4EBP (p4EBP). We found that over-expressing zfh2 in EBs using the Su(H)GBEGal4ts driver is sufficient to detect increased p4EBP levels by immunostaining (Fig 5A). Similar induction was also detected in ISCs and EBs when UAS-zfh2EAB expression was driven by the esgGal4ts driver (S4A Fig). Over-expression of Tor activator Rheb or knock-down of Thor/4-EBP confirmed the specificity of the p4EBP signal (Figs 5A, S4A and S4B).

We next asked whether Tor signaling is essential for zfh2-mediated EB-growth. To this end, we inhibited the activity of Tor genetically, by over-expressing the Tor inhibitor complex TSC1 and TSC2, and pharmacologically, by feeding flies the Tor inhibitor rapamycin. Both

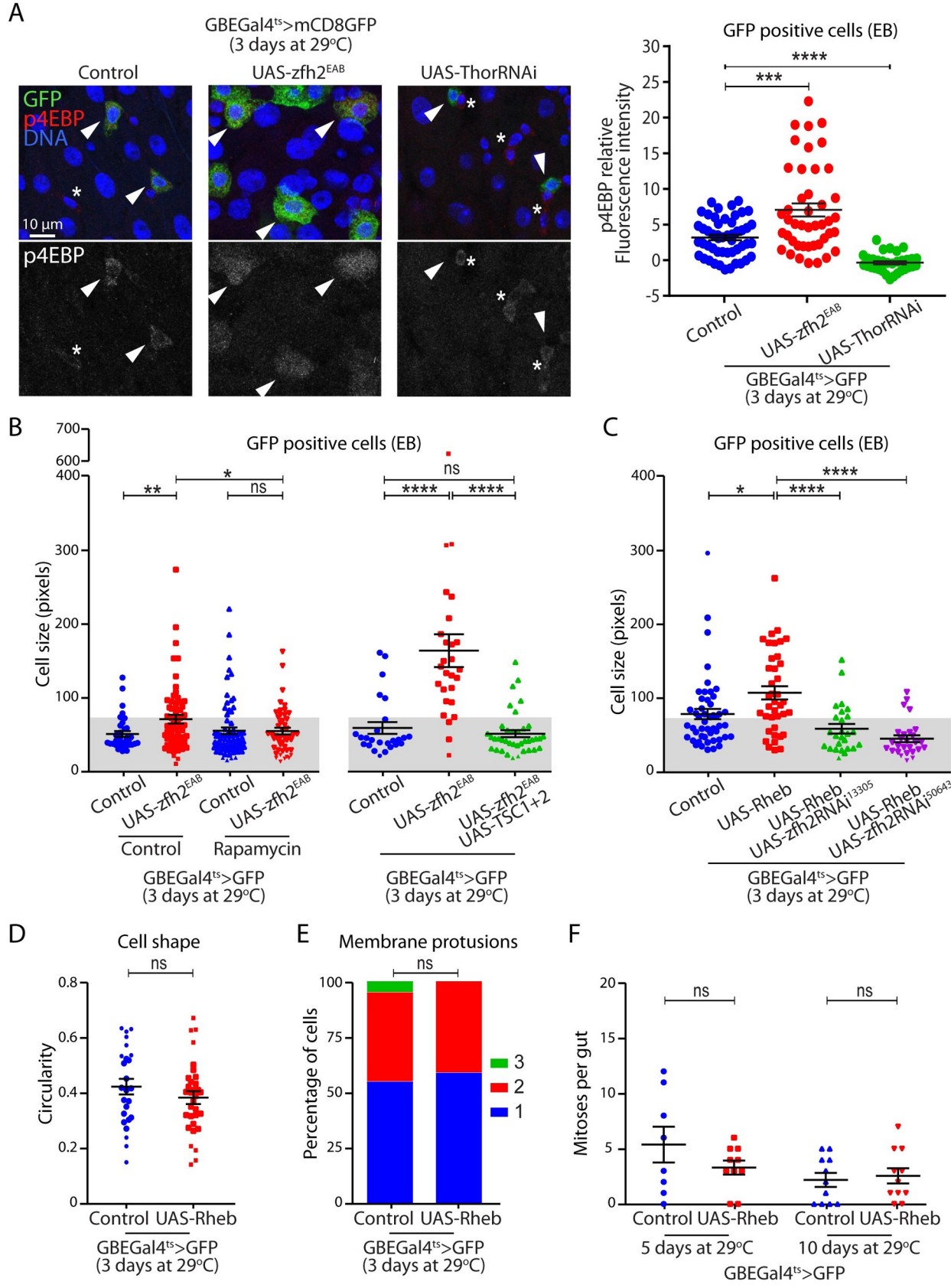

**Fig 5. zfh2-mediated activation is required for cell growth in enteroblasts.** (A) zfh2 is over-expressed by driving the UAS-zfh2EAB transgene using GBEGal4ts. 4EBP (Thor) is knocked-down in EBs by driving dsRNA using GBEGal4ts. EBs are labeled by GBEGal4ts> mCD8GFP and p4EBP is labeled via immunohistochemistry. p4EBP is detected in EBs (GFP positive cells) and in some small GFP negative cells (arrowhead and asterisk respectively). Protein levels are quantified by measuring mean fluorescence intensity of individual EBs. Inducing EB activation via zfh2 over-expression is sufficient to increase Tor signaling activity. (B) zfh2 is over-expressed by driving the zfh2EAB transgene using GBEGal4ts, Tor activity is inhibited via Rapamycin or by co-expression of the TSC1+2 complex. EBs are labeled by GBEGal4ts> mCD8GFP.Cell size of EB are quantified by measuring cell area of individual cells. Tor signaling is required for zfh2 mediated EB growth. (C) zfh2 is knocked-down in EB by driving dsRNA against zfh2 using GBEGal4ts, Tor activity is stimulated by over-expression of the Tor activator Rheb. Cell size of EB is quantified by measuring cell area of individual cells. Stimulation of Tor activity is sufficient to induce EB growth in a zfh2 dependent manner. Driving EB Growth via over-expression of the Tor activator Rheb is not sufficient to induce changes in cell morphology, measured by cell circularity (D), formation of membrane protrusions (E) or to induce ISC proliferation non-cell autonomously (F). In A, B, C, D, F, values are presented as average +/- s.e.m, and p-values are calculated using a two-tailed Student's t-test. In E p-values are calculated using the Mann-Whitney test.

manipulations are sufficient to block zfh2-mediated EB growth (Fig 5B), strongly suggesting that Tor signaling acts downstream genetically or temporally of zfh2-induced activation. To further test this hypothesis, we activated Tor signaling through the EB-specific over-expression of Rheb. Similar to what we observed for zfh2, this results in a robust EB growth but no significant increase in nuclear size (Figs 5C and S4C), suggesting that in EBs cell growth can be genetically separated from endoreplication. Remarkably, we found that co-expression of zfh2RNAi constructs is sufficient to block Rheb-induced growth (Fig 5C). Furthermore, we observed that Rheb expression and its associated EB growth fail to induce morphological changes, protrusion formation and ISC proliferation (Fig 5D, 5E and 5F), strongly suggesting that Tor signaling control cell growth but does not control EB activation.

The Ras/MAPK pathway has been recently shown to also control EB growth, more specifically under tissue damage conditions [12]. Similar to what we observed with Rheb over-expression, we found that although activation ERK signaling by expressing the active form Rolled$^{SEM}$ cause a detectable EB growth (S5A Fig). However, it does not significantly induce changes in EB morphology, protrusions or affect ISC proliferation rates (S5B, S5C and S5D Fig). In addition, although expressing a dominant form of Ras (Ras$^{N17}$) in EBs is sufficient to significantly reduce zfh2-mediated EB growth, it does not affect cell activation, measured by cell circularity and protrusions (S5E, S5F and S5G Fig).

Altogether, these data greatly improve our understanding of the sequence of event leading to EB differentiation. We show that EB activation can be separated from growth pathways and that zfh2-mediated activation is essential to prime EBs for the Tor and Ras/MAPK pathways to promote cell growth.

## zfh2 and insulin signaling act in parallel to control a subset of phenotypes associated with EB activation

The insulin signaling pathway controls ISC proliferation and EB growth [9, 10, 34, 35]. Particularly relevant to our study, insulin signaling is known to control very early EB growth and the degradation of E-cadherin adhesion between EBs and their neighboring stem cells, regulating ISC in a non-cell autonomous manner [11]. This suggested to us that, like zfh2, this pathway may act during the dormant to activated EB transition and could regulate the process of EB activation itself. Therefore, we investigated the genetic interaction between InR signaling and zfh2 in EBs. As reported previously, activation of the insulin signaling pathway in EB, by over-expression of an activate form of the insulin receptor (InRact) is sufficient to induce EB growth and drive ISC proliferation (Fig 6A and 6B). However, we found no evidence that this manipulation induces EB morphological changes or the formation of cell protrusions (Fig 6C and 6D), suggesting that at least some aspects of EB activation that are controlled by zfh2 are InR-independent. To confirm this notion, we tested the requirement for the insulin pathway in

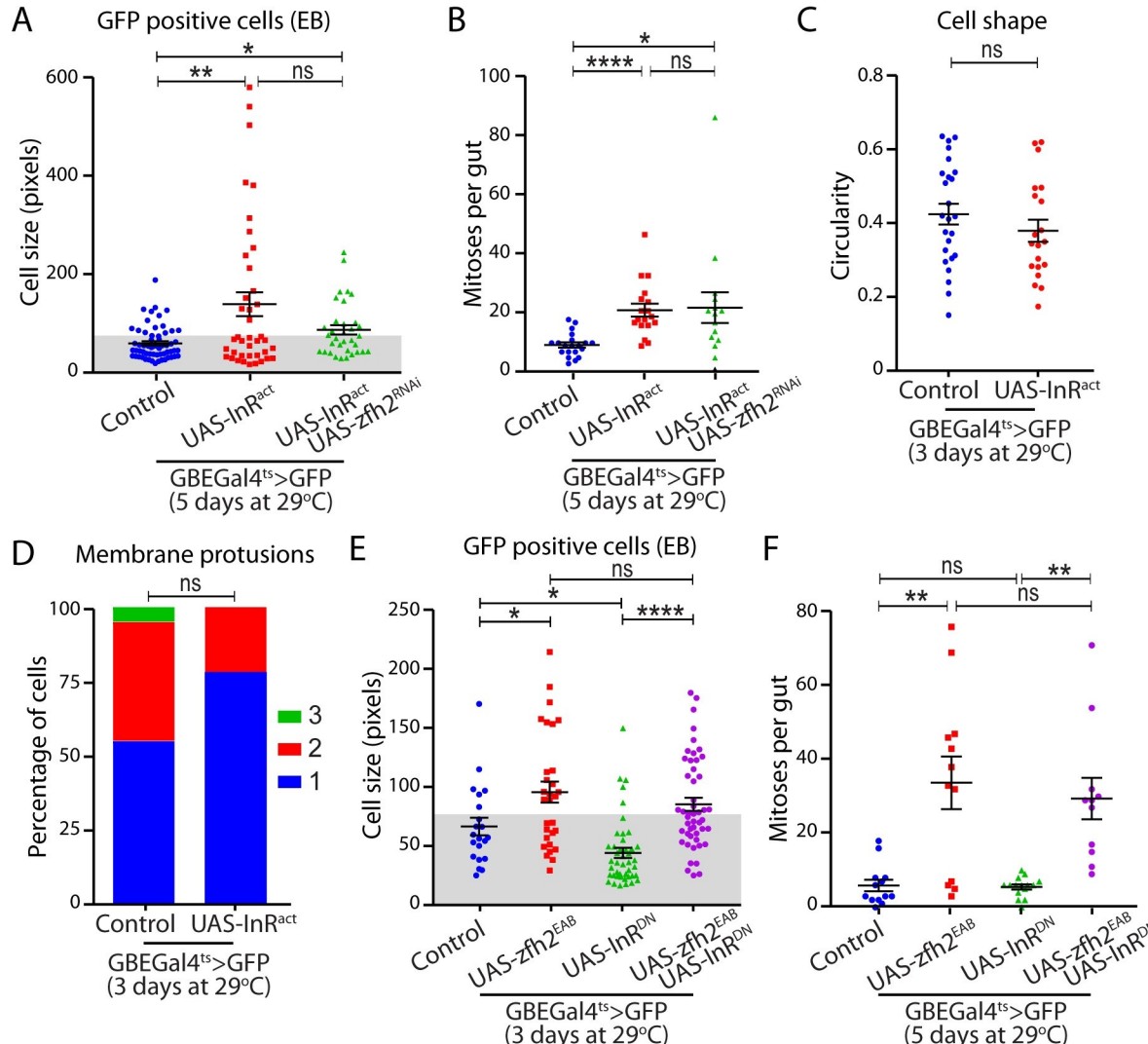

**Fig 6. zfh2 and insulin signaling act in parallel to control enteroblast activation.** (A, B, C, D) insulin receptor activity is induced in EBs by driving expression of the activated form of the insulin receptor (InRAct) using GBEGal4 ts. EBs are labeled by GBEGal4ts> mCD8GFP. (A) zfh2 is knocked-down in EBs by driving dsRNA against zfh2 using GBEGal4ts. Cell size of EBs is quantified by measuring cell area of individual cells. zfh2 is not required for InR-mediated EB growth. (B) zfh2 is knocked-down in EBs by driving dsRNA against zfh2 using GBEGal4ts. Mitoses per gut are detected via immunohistochemistry against phosphoHistone H3. Insulin receptor activity in EBs induces ISC proliferation non-cell autonomously, bypassing the requirement for zfh2-mediated activation. (C, D) Inducing insulin receptor activity in EBs is not sufficient to induce changes in cell morphology, measured by cell circularity (C), or formation of membrane protrusions (D). (E, F) Insulin receptor activity is blocked in EBs by driving expression of the dominant negative form of the insulin receptor (InRDN) using GBEGal4ts. zfh2 is over-expressed by driving the zfh2EAB transgene using GBEGal4ts. Cell size of EB are quantified by measuring cell area of individual cells (E). Mitoses per gut are detected via immunohistochemistry against phosphoHistone H3 (F). Blocking insulin receptor activity in EBs blocks cell growth and ISC proliferation. Over-expression zfh2 rescues these phenotypes. In A,B,E,F values are presented as average +/- s.e.m, and p-values are calculated using a two-tailed Student's t-test. In D p-values are calculated using the Mann-Whitney test.

zfh2-induced activation. We found that, blocking insulin signaling by expression of a strong dominant negative form of InR does not affect zfh2 ability to induce EB growth and non-cell autonomous ISC proliferation (Fig 6E and 6F). This further suggest that zfh2 over-expression is sufficient to bypass the requirement of InR for EB growth that has previously described [11]. Finally, we performed the converse epistasis assay by activating insulin signaling in a

background where zfh2 expression was decreased by dsRNA. In these conditions, we found that zfh2 is dispensable for InR-induced ISC proliferation (Fig 6C) and we detected a small but not significant decrease in EB cell size (Fig 6A), suggesting that zfh2 does not function genetically downstream of the InR signaling cascade to regulate EB growth and the subsequent non-cell autonomous ISC proliferation.

Together, these genetic interactions demonstrate that InR signaling can promote growth in zfh2-deficient EBs and that, conversely, zfh2 can induce growth in the absence of InR activity. However, InR is not sufficient to induce protrusion or morphological changes. Therefore, our work strongly suggests that InR and zfh2 act in parallel to control some aspects of early EB growth, but that zfh2 has a unique role in the migratory phenotype associated with EB activation.

### Maintaining zfh2 expression prevents enterocyte terminal differentiation

During our zfh2 gain-of-function experiments, we noticed that prolonged over-expression of zfh2 using the esgGal4ts drivers leads to dramatic disruption of the intestinal architecture. While the wild-type young intestinal epithelium is invariably organized in a monolayer, we observed the formation of multilayered cell clusters at day 3 of zfh2 induction. This ultimately leads to intestines where the epithelium is so overgrown that the lumen becomes undetectable by day 10 (Fig 7A). This phenotype is remarkable in the anterior and the posterior midgut (S6C Fig), but limited in the copper cells region (mid midgut) where proliferation is often reduced and zfh2-expressing tumors are much smaller. Importantly, we found that in both posterior and anterior midgut, cell division is limited to basal layers of the tumor (S6C Fig), suggesting that the tumor phenotype is driven by proliferation of ISCs.

Multilayered tumors comprised of adult intestinal progenitors have been reported when terminal differentiation of EBs into the ECs is blocked [14, 15]. To test whether the cells that accumulate in these tumor-like epithelia are indeed EBs and what stage of their differentiation may be arrested, we used a combination of ISC, EE, EB and EC markers. We found that in these overgrown epithelia only a few cells in the basal layer express the ISC marker Delta or the EE marker Prospero (Fig 7B). Importantly, we found that in both posterior and anterior midgut, cell division is limited to basal layers of the tumor (S6C Fig), suggesting that the tumor phenotype is driven by proliferation of these ISCs. To confirm that most tumor cells are EBs, we investigated later differentiation markers. We found that, in addition to escargot, they expressed high levels of Sox21a and are negative for the EC marker Pdm1 (Fig 7C and 7D). Finally, clonal analysis confirmed that maintaining zfh2 expression in intestinal lineages block terminal differentiation, as shown by the large reduction in the number of Pdm1-positive cells in zfh2-overexpressing MARCM clones compared to control clones (S6 Fig).

Expression of Notch activity reporters, such as the Su(H)GBE-LacZ and Su(H)GBEGal4 lines, has been used as an historical marker of EBs [30]. Interestingly, in the multilayered intestines of esgGal4ts>zfh2 animals, we observed several layers of GBE-positive cells basally. However, the many apical layers of these tumor-like epithelia are composed exclusively of GBE-negative esg-positive cells (Fig 7E). In addition, multilayered cell clusters form when zfh2 is specifically over-expressed in EBs using the GBEGal4ts driver, although at a slightly slower pace and not to the same magnitude that when using the esgGal4ts driver (Fig 7F). This confirms that this phenotype is cell autonomous and suggests that, like the GBE-LacZ reporter, the activity of GBEGal4 driver is lost in late differentiated EBs, resulting in a transient zfh2 over-expression and weaker phenotype in GBEGal4ts>UAS-zfh2$^{EAB}$ animals. Together, these data demonstrate that ectopically maintained zfh2 expression blocks terminal differentiation into ECs and arrest EBs in a previously uncharacterized esg-positive Sox21a-high Notch-negative cell state.

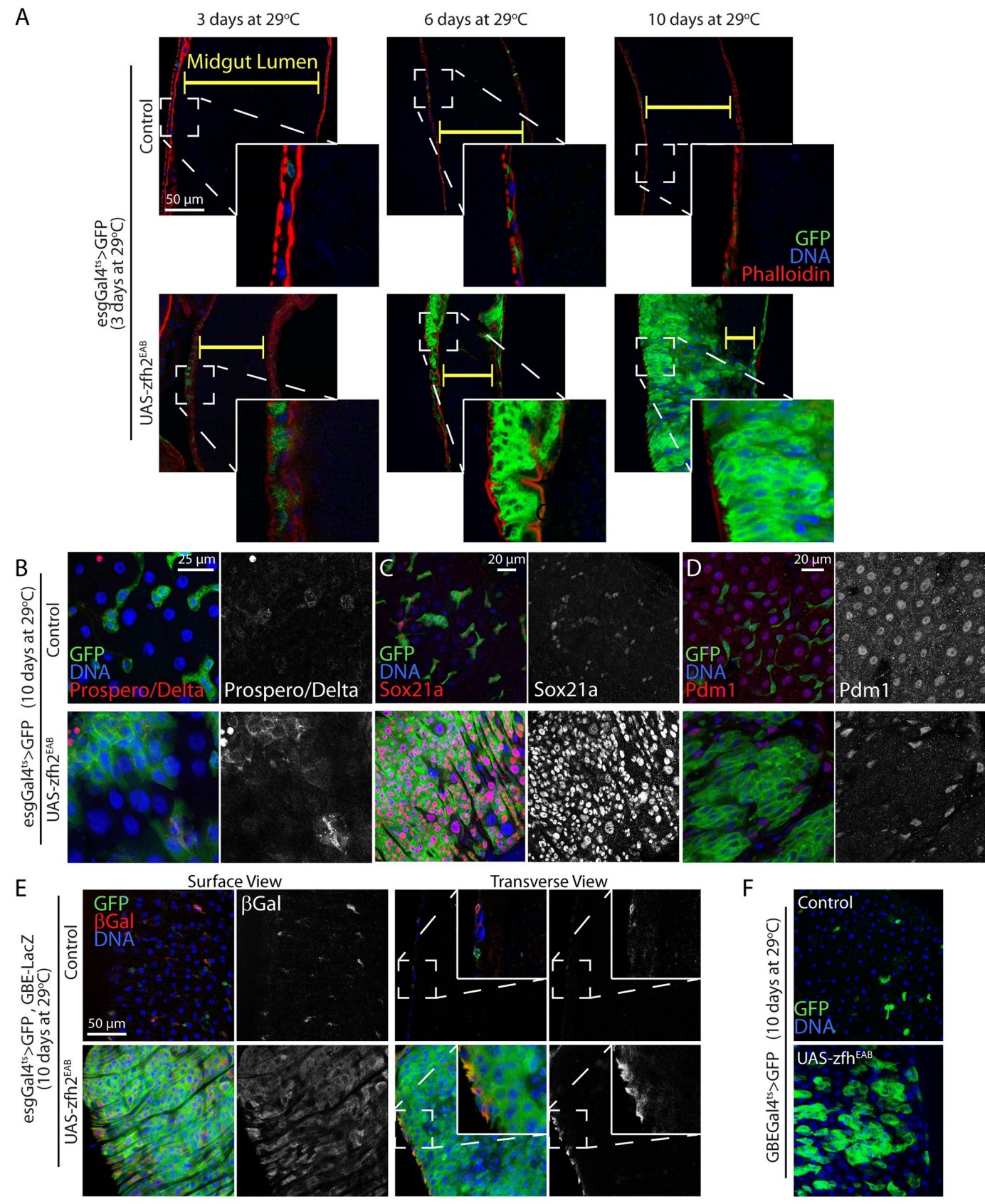

**Fig 7. Maintaining zfh2 expression prevents enterocyte terminal differentiation.** (A,B,C,D,E) Representative confocal image of the posterior midgut. zfh2 is over-expressed by driving the UAS-zfh2EAB transgene using esgGal4ts. (A) Transverse view of the posterior midgut. Visceral muscle is stained via phalloidin conjugated with Cy5. Long term zfh2 over-expression leads to multilayered escargot expressing tumors. (B) ISCs and EEs are labeled via immunohistochemistry against delta (membrane signal) and prospero (nuclear signal) respectively. Limited number of cells within the zfh2 over-expression tumors are ISCs or EEs. (C) ISCs and EBs are labeled via immunohistochemistry against Sox21a. zfh2 over-expression tumor cells show high levels of sox21a protein. (D) ECs are labeled via immunohistochemistry against Pdm1. zfh2 over-expression tumor cells are Pdm1-negative. (E) βGalactosidase, from the EB-specific reporter Su(H)GBELacZ, is detected via immunohistochemistry. Most of the cells located in basal layers of the zfh2 over-expressing tumors are positive for this reporter. However, transverse view shows that only the basal layers of these tumors are LacZ positive. (F) Representative confocal image of the posterior midgut. zfh2 is over-expressed by driving the UAS-zfh2EAB transgene using GBEGal4ts. EBs are labeled via GBEGal4ts>GFP. Long term EB-specific zfh2 over-expression leads to accumulation of EBs.

## Discussion

### zfh2 and enteroblast activation

In this study, we characterized the stepwise series of events leading to intestinal progenitor differentiation and the role of the transcription factor zfh2 in early progenitor activation.

Previous work has shown that challenging homeostatic conditions, either via bacterial infection or chemical stressors, is sufficient to induce phenotypical changes in EBs leading to terminal differentiation into the EC fate [13]. The changes associated with early EB activation have been described as Epithelial-Mesenchymal Transition (EMT)-like, as they are coupled with re-organization of the cytoskeleton and migratory behavior. Our work establishes a sequential order and gives temporal resolution for some of these events (Fig 8). We show that the morphological changes, associated with EB migration, and the formation of thin membrane protrusions, phenotypes that we describe as being part of the activation process, take place within 6 hours of tissue damage. Interestingly, while morphological changes persist, suggesting that EBs remain migratory, thin protrusions are lost 12 hours after induction suggesting a shift in signaling (Fig 8A). It will be interesting to ask whether cell surface receptors or signaling ligands transiently accumulate in these structures, as they do in other stem cell compartment and developing tissues [36], and test the possibility that EBs use these cellular processes to determine migration orientation relative to sites of tissue damage. In addition to EBs, it may also be interesting to test whether similar processes are involved in early cell differentiation in the Prospero-positive endocrine lineage.

In parallel to our improved description of the early activation process, previous work has shown that EB early differentiation indirectly affects ISC proliferation (Fig 8B). Small dormant EBs inhibit proliferation of their neighboring ISC in a contact dependent manner via high levels of DE-Cadherin; degradation of DE-cadherin is controlled by the insulin signaling pathway in EBs to promote ISC proliferation [11]. In addition, blocking terminal differentiation induces non-cell autonomous ISC proliferation and tumorigenesis, through the activation of several pathway including JAK/STAT and EGF [14, 15, 32]. Here we found that zfh2-mediated EB activation is sufficient to promote ISC proliferation non-cell autonomously and our data suggest that this induction doesn't require the activity of the insulin receptor. We also show that InR signaling is not sufficient to promote EB activation, defined by morphological changes and the formation of membrane protrusions. Together, these observations support a model where both InR and zfh2 can induce independently EB growth and ISC proliferation but where zfh2 is uniquely required for the migratory EB phenotype. Unfortunately, at this point, it is unclear how InR activity controls to E-cadherin degradation. It would be interesting in future studies to test whether zfh2 regulates ISC proliferation non-cell autonomously through similar mechanisms, once these have been identified. The function of zfh2 would guarantee that EB migration and early growth are coordinated with its detachment from the neighboring stem cell and compensatory ISC proliferation to maintain tissue homeostasis.

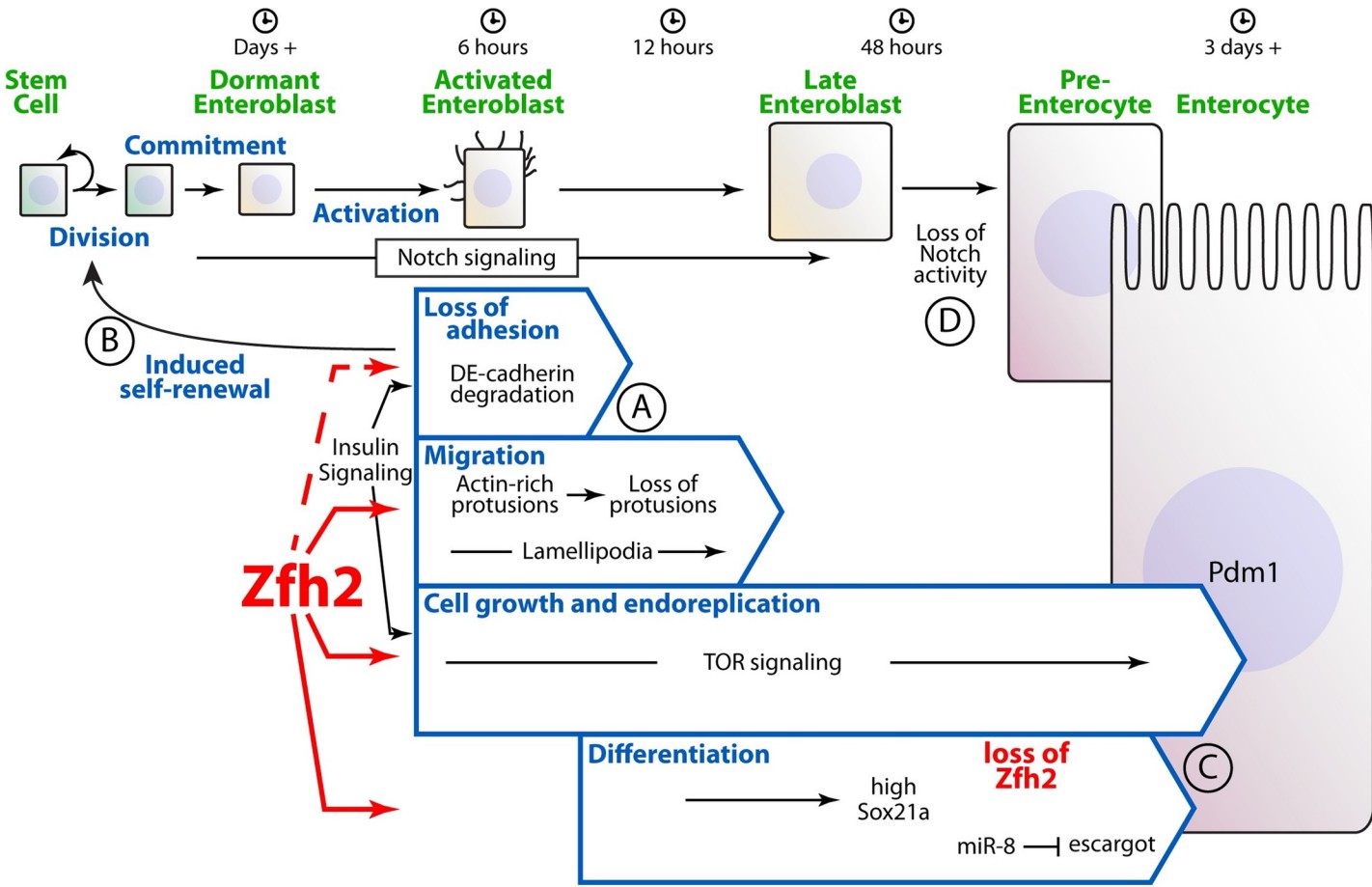

**Fig 8. Model presenting the function of zfh2 and the processes of enteroblast activation and differentiation.** (A) zfh2 controls the activation of EBs, as shown by the formation of thin membrane protrusion and morphological changes, 6 hours after stress exposure. Protrusions are lost 12 hours after activation. (B) EB activation and loss of cell adhesion induces compensatory stem cell proliferation. (C) Maintaining ectopic zfh2 activity in late EBs results in a block of differentiation, growth and tumor formation. (D) Notch signaling is inactive in zfh2 overexpression-induced tumors, suggesting that Notch activity is lost before esg down-regulation and other late differentiation processes.

Downstream of the initial process of activation, EB growth also requires zfh2-mediated activation. In addition, like InR signaling, while activation of TOR or MAPK pathways is sufficient to promote EB growth, it does not result in EB activation or ISC proliferative response. This strongly suggests that EB growth and growth-related signaling pathways are temporally and genetically downstream of zfh2-mediated activation.

## zfh2, enteroblast differentiation and tumorigenesis

On the other end of the differentiation process, several factors have been shown to regulate terminal differentiation of EBs into ECs. Here we show that maintaining zfh2 expression in late EBs prevents their differentiation and causes the formation of highly proliferative multilayered tumors, suggesting that zfh2 expression and activity need to be turned "off" for terminal differentiation (Fig 8C). Interestingly, non-proliferative tumorigenic EBs have also been reported when the transcription factor Sox21a is inhibited [14, 15, 32]. However, we show that overexpression of zfh2 is sufficient to increase and maintain high levels of Sox21a, suggesting that these two types of EB tumors differ. In addition, we show that Notch activity is lost in the most apical layers of cells that accumulate when zfh2 expression is sustained, whereas Notch activity

is maintained in Sox21a mutant tumors. Finally, zfh2-mediated tumors are more aggressive than most Sox21a mutant tumors, as they are become obvious by day 3 of transgene activation and lead to massive multilayering in both the anterior and posterior midgut by day 10. Interestingly, high levels of Sox21a are sufficient to drive EB differentiation into Pdm1-positve ECs [14, 15]. Therefore, our data support a model where zfh2-overexpressing EBs are arrested later in differentiation than Sox21-mutant EBs. This also suggests that in these late EBs, maintaining zfh2 expression directly or indirectly inhibits the ability of Sox21a to drive differentiation thus leading to a chronic expression of self-renewal factors. We anticipate that identifying the target genes of zfh2 will shed light on the molecular mechanisms of this interaction.

Our results also strongly suggest that Notch activity downregulation precedes escargot and Sox21a downregulation, likely when the EB migrates away from its neighboring stem cell thus losing the Delta signal required for EB commitment and early differentiation [30, 37] (Fig 8D). This further improves our understanding of the several cellular transitions taking place during the EB differentiation process and suggests that other intermediate phenotypes have yet to be identified in this lineage. Other genetic elements have been shown to regulate late EB differentiation events. The microRNA mir-8, that has been hypothesized to promote proper precursor epithelial reintegration and terminal differentiation by inhibiting expression of escargot [13, 38]. Our results show that long-term zfh2 over-expression induces maintenance of a mesenchymal-like state in late EBs, suggesting that mir-8 dependent terminal differentiation and reintegration are temporally downstream of zfh2 activity loss. Despite the recent advances of live imaging in the intestinal lineage [39], we do not have sufficient temporal resolution to visualize the sequence of events leading to EB terminal differentiation. Further experiments will be required to understand the exact relationships between these critical differentiation regulators.

Of note, another gene that has been recently implicated in maintenance of stemness in this tissue is the *Drosophila* ZEB protein homolog zfh1 [13]. Despite being similarly named, there is very low similarity (10.2%) between these two proteins: zfh1 has 1 homeodomain and 7 zinc-fingers, while zfh2 has 3 homeodomains and 17 zinc fingers. This differences in domain number and organization makes it very unlikely that these 2 proteins have overlapping target genes. However, it would be interesting to test possible genetic interactions between zfh2 and zfh1 in further studies.

Finally, our study uncovers the role of zfh2 in the EMT-like EB activation and MET-like EB terminal differentiation. In mammals, the zfh2 homolog ATBF1/ZFHX3 (AT-motif binding factor/zinc finger homeobox 3) is a demonstrated tumor suppressor, found frequently mutated in a large spectrum of human cancers (e.g. prostate cancer [22, 25]). However, the precise mechanism(s) by which ATBF1 controls tumorigenesis and metastasis remain largely elusive. More interestingly, in vitro studies correlate over-expression of another zfh2 homolog, ZFHX4, with cancer cell migration and invasiveness[27]. It is tempting to propose that further investigating the function of zfh2 in the *Drosophila* intestinal lineage will generate new hypotheses regarding the role of ATBF1 and ZFHX4 in tumor formation and progression.

## Mechanism and regulation of zfh2 activity

Our study demonstrates that zfh2 controls EB activation, growth and terminal differentiation. However, its direct target genes and how it controls, directly or indirectly, critical EB processes and signaling pathways remain unclear. While zfh2 regulates targets of Notch signaling during tarsal development [19], it is unlikely that it functions in a similar mechanism in EBs as knocking-down its expression does not affect Notch reporters activity. Our work has shown that zfh2 mediated activation is sufficient to induce Tor activity, and that this increase of Tor activity leads to EB growth. In EBs, Tor signaling controls cell growth via the TORC1 complex [9]

and is regulated by Notch signaling through the regulation of TSC2 expression [10]. However, we don't know how the transcription factor zfh2 may regulate the TORC1 signaling cascade, but it is likely a secondary effect of EB activation rather than a direct regulation. We have further shown that zfh2 over-expression is sufficient to drive a host of phenotypes, including formation of membrane protrusions and changes in cell shape, suggesting that zfh2 mediated activation leads to a remodeling of the cytoskeleton. Again, the mechanisms by which zfh2 controls EB morphology and potentially migration remain unknown.

Finally, here we show that zfh2 is expressed in both ISCs and EBs but we focused on the role that zfh2 has in EB activation and differentiation. Thus far, we have found no phenotype associated with zfh2 loss in ISCs. The mechanism by which zfh2 activity is regulated in an EB specific manner remains to be elucidated. Activity of ATBF1, one of the zfh2 mammalian homolog, has been shown to be controlled by its subcellular localization [40, 41], however we didn't observe any difference in either expression level or cellular localization of zfh2 between ISCs and EBs. Alternatively, zfh2 may be regulated by post-translational modifications; for example, ATBF1 has been shown to be SUMOylated and also phosphorylated [41, 42]. Another possible mechanism for gene target specificity in EB could be combinatorial regulation, either via direct or indirect interaction with other transcription factors. Biochemical analysis of zfh2 and its potential partners in the adult intestine will be required to make significant progress in our understanding of the regulation of zfh2 activity.

## Materials and methods

### Drosophila stocks and rearing

All flies were raised on standard yeast and molasses-based food, at 25˚C and 65% humidity, on a 12 h light/dark cycle. For TARGET experiments flies were raised at 18˚C and shifted to 29˚C 4–7 days after eclosion.

The following strains were obtained from Bloomington *Drosophila* stock center: w1118, UAS-zfh2RNAi50643 (50643), UAS-zfh2EAB (56545), zfh2Gal4GMR73G11 (81071), esgLacZ (10359), UAS-Gtrace (28280), UAS-Rheb (9688), UAS-InRwt (8262), UAS-InRDN (8252), UAS-InRAct (8263), UAS-mCD8RFP (27391), UAS-MoesinGFP (31775), UAS-RasN17 (4845), UAS-ThorRNAi (80427). From Vienna *Drosophila* RNAi Center: UAS-zfh2RNAi 13305 (v13305). The following strains were gifts from: Su(H)GBELacZ by S.Bray, esgGal4ts,Su(H)GBEGal80 by H. Jasper, DeltaGal4 and Su(H)GBEGal4 by S.X.Hou, UAS-TSC1+TSC2 by M. Tatar, ssg-Gal4 by S. Hayashi, UAS-Rolled SEM by M. Mlodzik, UAS-zfh2RNAiSp and zfh2Gal4LP30 by FJ Díaz-Benjumea.

For each of the described experiments, the genotype of the flies used is detailed in S1 File.

### Immunohistochemistry and imaging

Female fly intestines were dissected in PBS 1X solution and then fixed in Fixation Buffer containing 100 mM glutamic acid, 25 mM KCl, 20 mM MgSO4, 4 mM sodium phosphate, 1 mM MgCl2, and 4% formaldehyde for 45 minutes at room temperature. Samples were blocked using Blocking buffer containing PBS 1X, 0.5% BSA, and 0.1% Triton X-100. Samples were then incubated in the same buffer containing primary antibodies over-night at 4˚C [anti-phospho-Histone H3 from Millipore (1:2000), anti-Beta-Galactosidase from DHSB (1:500), anti-Prospero from DHSB (1:200)]. Fluorescent secondary antibodies were obtained from Jackson Immunoresearch. DNA was stained using Hoechst and Visceral muscle was stained using Alexa Fluor 647 Phalloidin (from Invitrogen, 1:400). For Delta (from DHSB, 1:500), Sox21a (generated in the lab [43], 1:5000), p4EBP (from Cell signaling technologies, 1:200), zfh2 (kindly provided by Chris Doe, 1:200) and PDM1 (kindly provided by Xiaohang Yang and Cai

Yu, 1:1000) staining samples were fixed in Fixation Buffer and Heptane, dehydrated with 100% Methanol and progressively re-hydrated in blocking Buffer. Confocal imaging was done using the Leica SP5 system and processed with Adobe Photoshop CS6.

### Imaging and analysis of non-fixed samples

Samples were dissected in 1XPBS solution, immediately mounted in Voltalef 10S oil (VWR). Confocal imaging was done using the Leica SP5 system and completed within 30 min of dissection. For analysis of cell circularity images were treated using FIJI. A composite image was created for each ROI, a Gaussian Blur Filter was used to remove noise and remove membrane protrusions. Images were then converted to Binary before measuring circularity using FIJI. For analysis of membrane protrusions untreated z-stacks were manually analyzed.

In all quantification of cell area, cell morphology and membrane protrusions, at least 10 ROI were analyzed for each condition from at least 4 different female flies. Numerical values are presented in S2 and S3 Files.

### Quantification of protein levels by immunofluorescence intensity

To quantify fluorescence intensity, images were analyzed using FIJI [44]. For each EB or ISC GFP-positive cell, a single focal plane through the center of the cell was selected. First, the GFP signal was used to delineate individual cell area. The mean fluorescence intensity of pixels for the antibody staining channel was measured within the cell area. This value was then normalized to mean background pixel intensity in a comparable neighboring region of interest selected between GFP cells. The data are presented as normalized pixel intensity within GFP-positive cells. Numerical values are presented in S2 and S3 Files.

### Western blot analysis of proteins

Female fly intestines were dissected in PBS 1X. Proteins were extracted in Laemmli sample buffer, separated on a 15% SDS poly-acrylamide gel and transferred according to standard procedures. Antibodies directed against B-actin (Cell signaling technologies, 1:5000) and p4EBP (Cell signaling technologies, 1:1000) were used.

### Rapamycin treatment

Rapamycin (Fisher Scientific) was dissolved in DMSO and added to the food at a final concentration of 200μM. 4–7 days old flies were shifted to food containing either DMSO or DMSO and Rapamycin for 4 days before being shifted to 29˚C for 3 days. Food with either DMSO or DMSO and Rapamycin was changed every 2 days.

### DSS, Paraquat and Ecc15 treatments

For DSS stress exposure, 4–7 days old flies were starved for 6 hours in empty vials before being transferred to vials containing a filter paper with either 5% Sucrose (AMRESCO) or 5% Sucrose and 4% DSS (Sigma Aldrich). For Paraquat stress exposure flies were similarly starved before being transferred to vial containing either 5% Sucrose or 5 mM Paraquat dissolved in 5% Sucrose. Erwinia carotovora carotovora15 (Ecc15) was grown overnight in LB medium at 30˚C and harvested by centrifugation. The bacterial pellet was then resuspended in a 5% Sucrose solution before being fed to 6 hours starved flies via filter paper. For acute stress flies were dissected at the indicated times.

## Mosaic analysis with a repressible cell marker clones (MARCM) and lineage tracing

Clones were generated using the following MARCM stock: MARCM19 (hsFLP, tubGal80, FRT19A; T80Gal4, UAS-GFP). 4–7 days old flies were heat-shocked at 37˚C for 30 min for clone induction, and then transferred at 25˚C for 7 days before dissection and immunohistochemistry.

Lineage tracing used the GTRACE system (w*;UAS-RedStinger,UAS-FLP,UbiFRTStop FRTStinger) combined with the temperature sensitive driver GBEGal4ts. 4–7 days old flies were shifted at 29˚C for 3 days to allow for transgene expression. Flies were then starved for 6 hours in empty vials before being transferred for 24 hours to vials containing either 5% Sucrose or 5% Sucrose and 4% DSS. Flies were shifted back to regular food for 2 days before dissection and immunohistochemistry.

## Supporting information

**S1 Fig. zfh2 is expressed in ISCs and EBs.** (A) Representative confocal image of the posterior midgut. EB are labeled by GBEGal4ts>mCD8GFP. Sox21a and zfh2 are detected via immuno-histochemistry in ISC and EB. Flies are fed either DSS or Sucrose for 6, 18 or 40 hours before dissection and fixation. zfh2 protein is detected via immunohistochemistry. (B) zfh2 protein levels in EB are measured by quantification of zfh2 fluorescence in individual cells. Values are normalized to neighboring ISC. zfh2 protein is expressed at similar levels in both ISC and EB. (C) zfh2 protein levels in EB and ISC are measured by quantification of zfh2 fluorescence in individual cells. zfh2 protein levels increase after DSS mediated stress. In B and C, values are presented as average +/- s.e.m, and p-values are calculated using a two-tailed Student's t-test. (TIF)

**S2 Fig. zfh2 does not controls intestinal cell composition but regulates EB cell size.** (A) ISC and EB are labeled by esgGal4ts>GFP. ISC and enteroendocrine cells are labeled via immuno-histochemistry against delta and prospero respectively. zfh2 is knocked-down by driving dsRNA against zfh2 using esgGal4ts. Number of ISC (GFP+,Delta+), EB (GFP+,Delta-) and ee (Prospero+) cells are quantified and normalized to the total number of cells per ROI. Each value represents a ROI. (B) ISC and EB are labeled by esgGal4ts > GFP. ISC and enteroendo-crine cells are labeled via immunohistochemistry against delta and prospero respectively. zfh2 is over-expressed by driving the UAS-zfh2EAB transgene using esgGal4ts. zfh2 is knocked-down by driving dsRNA against zfh2 using esgGal4ts. Nuclear size of ISCs and EBs are quanti-fied by measuring nuclear area of individual cells. zfh2 knock down via dsRNA blocks endore-plication in EBs. (C) EB are labeled by GBEGal4ts>mCD8GFP. zfh2 is over-expressed by driving the UAS-zfh2EAB transgene using GBEGal4ts. zfh2 is knocked-down by driving dsRNA against zfh2 using GBEGal4ts. Nuclear size of EBs are quantified by measuring nuclear area of individual cells. zfh2 knock down via dsRNA blocks endoreplication in EBs. In A, B and C, values are presented as average +/- s.e.m, and p-values are calculated using a two-tailed Student's t-test. (TIF)

**S3 Fig. Stress- and zfh2-mediated induction of EB activation.** (A) Representative confocal images of non-fixed posterior midguts. EB are labeled by GBEGal4>mcD8GFP. Stress medi-ated EB activation is induced by feeding flies Paraquat or Ecc15 for 3–4 hours. Paraquat and ECC15 mediated stress is sufficient to increase the number of EBs with membrane protrusions (B) and decrease circularity (C). (D) Representative confocal images of non-fixed posterior midguts. EBs are labeled by GBEGal4>mcD8RFP, actin is labeled by GBEGal4>Moesin-GFP.

Stress mediated EB activation is induced by DSS for 6 hours. Membrane protrusions contain actin. (E) Representative confocal images of posterior midguts. EB are labeled by GBE-Gal4ts>GFP. zfh2 is over-expressed by driving the UAS-zfh2EAB transgene using GBEGal4ts. Sox21a is detected via immunohistochemistry. (F) Quantification of sox21a protein levels in EB by quantifying mean sox21a fluorescence levels in individual cells. zfh2 over-expression in EB increases sox21a levels. In C and F values are presented as average +/- s.e.m, and p-values are calculated using a two-tailed Student's t-test. In B p-values are calculated using the Mann-Whitney test.
(TIF)

**S4 Fig. zh2 over-expression induces TOR activity.** (A) zfh2 is over-expressed by driving the UAS-zfh2EAB transgene using esgGal4 ts. 4EBP (Thor) is knocked-down in EB by driving dsRNA using EsgGal4ts. Tor activity is stimulated by over-expression of the Tor activator Rheb. p4EBP is labeled via immunohistochemistry. (B) Protein levels are quantified by measuring mean fluorescence intensity of individual cells. Inducing EB activation via zfh2 over-expression is sufficient to increase Tor signaling activity. (C) EB are labeled by GBEGal4ts > mcD8GFP. zfh2 is over-expressed by driving UAS-zfh2EAB using GBEGal4ts. Tor activity is induced by over-expressing Rheb using GBEGal4ts. Nuclear size of EB are quantified by measuring nuclear area of individual cells. In B and C values are presented as average +/- s.e.m, and p-values are calculated using a two-tailed Student's t-test.
(TIF)

**S5 Fig. Interaction between zfh2 and the Ras/MAPK pathway.** (A,B,C,D) ERK activity is induced in EB by driving the expression of the activated form of ERK (RolledSEM) using GBE-Gal4ts. EB are labeled by GBEGal4ts> mCD8GFP. (A) Cell size of EB are quantified by measuring cell area of individual cells. ERK activity induces EB growth. Inducing ERK activity is not sufficient to induce changes in cell morphology, measured by cell circularity (B), an increase on mitoses per gut, detected via immunohistochemistry against phosphoHistone H3 (C), or formation of membrane protrusions (D). (E,F,G) Ras activity is blocked in EB by driving expression of the dominant negative form of Ras (RasN17) using GBEGal4ts. zfh2 is over-expressed by driving the zfh2EAB transgene using GBEGal4ts. (E) Cell size of EB are quantified by measuring cell area of individual cells. Blocking Ras activity blocks EB growth cell-autonomously. Inducing EB activation induces growth in RasN17 EB. Blocking Ras activity is not sufficient to block changes in cell morphology, measured by cell circularity (F) or formation of membrane protrusions (G) associated with zfh2 mediated EB activation. In A, B, C, E, F values are presented as average +/- s.e.m, and p-values are calculated using a two-tailed Student's t-test. In D, G p-values are calculated using the Mann-Whitney test.
(TIF)

**S6 Fig. Long-term over-expression of zfh2 blocks EB differentiation and promotes tumor formation.** (A) Representative confocal images of control and zfh2 over-expressing MARCM clones in the posterior midgut, 7 days after induction. ECs are labeled via immunohistochemistry against Pdm1. (B) Proportion of EC per clone is quantified showing that UAS-zfh2 EAB clones contain a significantly reduced the number of EC compared to controls. (C) Representative confocal images of the posterior and anterior midgut. zfh2 is over-expressed by driving UAS-zfh2EAB using esgGal4ts. Mitotic cells are identified via immunohistochemistry against phosphoHistone H3 (arrowhead). Long term zfh2 over-expression leads to multilayered escargot expressing tumors in both the posterior and the anterior midgut. A high number of mitotically active cells are present in the basal layer of the tumor. Gut lumen is indicated by yellow brackets in the transverse views. In B values are presented as average +/- s.e.m, and p-values

are calculated using a two-tailed Student's t-test.
(TIF)

**S1 File. Detailed genotypes.** For each figure (main figures and supplementary figures), the genotypes of all the animals used in the described experiments is detailed here.
(DOCX)

**S2 File. Numerical Values–Main figures.** The numerical values presented in each of the main figure (Figs 1–7) are reported here.
(XLSX)

**S3 File. Numerical Values–Supplementary figures.** The numerical values presented in each of the supplementary figures (S1–S6 Figs) are reported here.
(XLSX)

## Acknowledgments

We are thankful to Taylor McKenty for technical assistance. We would also like to thank H. Jasper, S. Bray, M. Tatar, S. Hayashi, D. Bohmann, M. Mlodzik, FJ. Diaz-Benjumea, the Bloomington *Drosophila* stock center and the Vienna *Drosophila* RNAi Center for providing fly lines essential to this work and the Developmental Hybridoma Studies Bank and Chris Doe, Xiaohang Yang and Cai Yu for providing critical antibodies.

## Author Contributions

**Conceptualization:** Sebastian E. Rojas Villa, Fanju W. Meng, Benoît Biteau.

**Funding acquisition:** Benoît Biteau.

**Investigation:** Sebastian E. Rojas Villa, Fanju W. Meng.

**Methodology:** Sebastian E. Rojas Villa.

**Supervision:** Benoît Biteau.

**Validation:** Benoît Biteau.

**Visualization:** Sebastian E. Rojas Villa, Benoît Biteau.

**Writing – original draft:** Sebastian E. Rojas Villa, Benoît Biteau.

**Writing – review & editing:** Sebastian E. Rojas Villa, Fanju W. Meng, Benoît Biteau.

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
