## [Decision Letter · Decision Letter 0]

16 Jul 2019

Dear Dr Biteau,

Thank you very much for submitting your Research Article entitled 'Zfh2 controls cell activation and differentiation in the adult Drosophila intestinal lineage' to PLOS Genetics. Your manuscript was fully evaluated at the editorial level and by independent peer reviewers. The reviewers appreciated the attention to an important problem, but raised concerns about the current manuscript. Based on the reviews, we will not be able to accept this version of the manuscript, but we would be willing to review again a much-revised version. We cannot, of course, promise publication at that time.

If you decide to revise the manuscript for further consideration at PLOS Genetics, please aim to resubmit within the next 60 days, unless it will take extra time to address the concerns of the reviewers, in which case we would appreciate an expected resubmission date by email to plosgenetics@plos.org.

[LINK]

We are sorry that we cannot be more positive about your manuscript at this stage. Please do not hesitate to contact us if you have any concerns or questions.

Yours sincerely,

Aurelio A Teleman

Associate Editor

PLOS Genetics

Gregory P. Copenhaver

Editor-in-Chief

PLOS Genetics

Reviewer's Responses to Questions

**Comments to the Authors:**

Reviewer #1: The authors report here that Zfh2 is an ISC/EB specific factor that regulates EB activation and differentiation to ECs. Furthermore, they report that Zfh2 is required and sufficient to drive activation of EBs. This includes cell shape change, cell size increase, transient formation of thin membrane protrusions and increased compensatory ISC proliferation. Additionally, they report that Zfh2 is upstream of TOR and MAPK and parallel to insulin signaling and that Zfh2 overexpression results in dysplastic lesions.

Overall this study explores an important question about the early stages of differentiation of intestinal progenitors- a process that is not well-understood. The data provided about Zfh2’s role in EB activation (i.e., cell shape and size change, protrusion formation, requirement for differentiation) are convincing; however, the mechanism(s) into how Zfh2 is regulated or regulates EB activation/differentiation needs to be strengthened. The claim that Zfh2 is upstream of TOR and MAPK and parallel to insulin signaling is not well supported with the data provided.

I have outlined my concerns below:

1. The role of Zfh2 needs to be put into context with other pathways known to regulate EB growth and differentiation, e.g. Notch or JAK-STAT signaling. The authors describe in the introduction that Notch controls Zfh2 expression in developing fly wing discs (page 6). Does the Notch signal in EBs promote Zfh2 expression or activity in EBs? This is important since the authors report that Zfh2 is required in EBs that have received a Notch signal (GBE+) for their growth and differentiation to ECs.

2. The study nicely tries to establish a standard to study EB activation and uses DSS stress to induce EB activation. It is important though to demonstrate that these activation events (i.e., protrusion formation) occur upon a variety of stresses shown to promote ISC proliferation (and thus likely increase activated EB numbers), e.g. pathogen infection, oxidative stress, DNA damage. Are these morphological changes common features of activation in other stress conditions or rather only a response to DSS treatment which affects the extracellular matrix. Antonello et al., 2015 reported EB protrusion formation only under non-stress conditions. Do EBs differentiating into EEs also form protrusions? Presumably, they also should.

3. The authors show that Zfh2 is required for cell size increase but is not sufficient to promote endoreplication. This data is confounding since the authors claim that Zfh2 also promotes TOR signaling which has been shown to promote endoreplication in fly tissues (Zielke et al., 2011).

4. In Figure 3D, why are there large protrusions observed in the control (without DSS)?

5. In Figure 4A, the overlap of Pdm1 with GBE>trace is difficult to observe. A separation of fields will help here. Is Zfh2 required for EE differentiation?

6. Do Zfh2 levels affect E-Cadherin levels in EBs? The text on page 23 states that Zfh2 can induce DE-Cadherin degradation, but this data is not in the manuscript.

7. The data shown to support TOR activation by Zfh2 is not yet convincing for the following reasons: 1) the phospho-4EBP antibody in Fig. 5A needs to be validated, preferably in situ (e.g., knockdown of 4EBP, overexpression of Rheb); 2) in Fig. 5B, Zfh2 overexpression increases phospho-4EBP in all esg+ cells (which presumably includes ISCs) in the field, yet there is no effect on ISC size or proliferation (Fig. 4C). This is surprising since it has been reported that activated ISCs can increase in size (Jiang et al., 2009); 3) in Fig. 5B, the activation of TOR signaling by Zfh2 overexpression is not terribly convincing, particularly since the background in the field shown is increased too. Furthermore, since Zfh2 overexpressing EBs are larger, the quantitation data needs to be normalized to cell volume; 4) the variation in phospho-4EBP levels on the western blots in Fig. S4A after Rheb overexpression is concerning as Rheb overexpression should give a consistent increase in TOR signaling. To support this it would be helpful to see phospho-4EBP levels in EBs overexpressing Rheb. For these reasons, it is not clear whether Zfh2 activates TOR signaling. Since TOR signaling is an important regulator of protein translation, cell growth and other cellular processes, it is also not terribly surprising that inhibiting TOR signaling blocks Zfh2 induced EB growth (Fig. 5C). Does Zfh2 loss in EBs increase TSC2 levels? (see Kapuria et al., 2012).

8. The abstract claims that Zfh2 is upstream of MAPK, but there is no data in the manuscript to support this conclusion. MAPK activity has been observed in ISCs (Jiang et al. 2011; Biteau and Jasper, 2011; Liang et al., 2017) but not EBs. The authors should show that MAPK activity is present in differentiating EBs under non-stress conditions. If so, is Zfh2 required for MAPK activation in EBs? Is MAPK (Rolled) required for Zfh2 to promote EB activation and differentiation?

9. In Fig. 6A, the authors show that overexpression of the activated insulin receptor promotes growth in Zfh2-deficient EBs. But this does not necessarily suggest that insulin signaling acts parallel to Zfh2 in promoting EB differentiation. The loss of TOR from EB blocks its growth and differentiation (similar to Zfh2 loss), but this can be bypassed during regeneration through activation of EGFR signaling (Jiang et al., 2016).

10. It’s not clear what Fig. 2A is showing even after reading the figure legend.

Some general comments:

1. The title could be more specific and state “progenitor cell activation“

2. Drosophila should be capitalized and in italics throughout the text

3. The manuscript would benefit (if known) from more information in the introduction about zfh2 and its regulation/activity in a variety of systems

Reviewer #2: In “Zfh2 controls cell activation and differentiation in the adult Drosophila intestinal lineage”, Villa, Meng and Biteau presents a very nice study of zfh2, a zinc-finger homeodomain transcription factor in regulating the activation of enteroblast (EB) in the adult fly. Under the physiological state, an EB can remain inactivated for up to 14days before differentiating into an enterocyte (EC). The molecular mechanism that confers EB with such long dormancy remains largely unexplored. Here Villa et al depicts zfh2 as a novel player in the determination of intestinal progenitor fate. By carefully tracking the morphological changes in EB coupled to loss- and gain-of-function approaches, the authors demonstrate a requirement for zfh2 for EB growth and activation induced by stress, yet maintaining ectopic zfh2 expression in activated EB blocks EB to EC differentiation, and eventually leads to tumor formation. Furthermore, by studying epistatic interaction between zfh and mTOR and insulin signaling, the authors are able to provide strong evidence that EB growth and morphological changes required for activation are largely separate processes.

While I like this study very much, I do have more questions regarding the data presented in Figure 7. The tumor phenotype of esg>zfh2 is convincing, striking and intriguing, and I would like to see a bit more characterization. First, are these tumors regionalized? For example, sox21a loss-of-function tumors preferentially grow in the anterior midgut (AMG) of the flies. Secondly, is the tumor formation driven by proliferation from ISCs or elevated endoreplication of the EB cells? It would be a nice control to characterize the tumors further with PH3 staining and BrdU incorporation, while labeling Zhf2 overexpressing cells. Lastly, the high levels of accumulation of Sox21a in the tumors seems counterintuitive based on what we know. It is the current understanding that Sox21a is necessary for EB differentiation into ECs, and sufficient to drive the expression of Pdm1 in the ECs; Loss of Sox21a results in tumor formation, and re-expression of Sox21a specifically in EB cells can prevent this process. The authors here show that overexpression of Zfh2 is sufficient to induce Sox21a expression, yet it correlates with tumor formation. The authors resolve the discrepancy by claiming that zfh2 overexpression tumors and Sox21a loss of function tumors are different: “Sox21a-negative esg-positive early enteroblast in sox21a loss of function and Sox21A-high esg-positive late enteroblast in zfh2 gain of function”. This explanation is vague and the language is awkward. If the authors mean that the sox-21a accumulating tumors are late EB cells, is it possible to examine mir8-lacZ expression in this tumors? In addition to attributing the difference to “early” and late” EBs, is it conceivable that Zfh2 overexpression blocks Sox21a activity, and the target of Sox21a is engaged in an ectopic amplification loop of Sox21a expression? It would be interesting to know what happens to Sox21a expression in a zfh1 loss of function clone. Also, what happens to Zfh2 expression in a Sox21a loss of function tumor, and also assess if there is also loss of NRE marker expression in the most apical layer.

Minor Issues:

1. The authors report an accumulation of EBs bearing small nuclei upon zfh2 knock-down. Based on this observation the authors propose that zfh2 is required for EB endoreplication. The authors should image endoreplication markers in WT and knock-down conditions to strengthen their conclusion.

2. The authors nicely show elevated p4EBP staining with UAS-zfh2 expression, but the corresponding Western blot in Fig5A and the three panels in FigureS4A are not so convincing. Is it possible to do a pS6K Western in the same condition? By the way, there is a curious observation in the b-actin blot: there are two bands in the UAS Zfh2 and UASRheb, but not in the controls. No I am not asking the authors to resolve the mystery, but if they happen to know why already, it would be nice to share in the figure legend.

3. The staining with Zfh2 antibody is faint in Figure S1, but am I correct to say that in the non-DSS treated guts, the staining is more diffuse, almost seems cytoplasmic. With DSS treatment, the nuclear staining becomes more prominent and discernable? This seems more obvious to me in the 18hr panel. It is difficult to draw such a conclusion from the figure presentation. It is also unlikely as in previous studies of Zfh2 expression pattern in the wing disc (Terriente et al, 2008), strong nuclear staining of Zfh2 is evident. However, if the authors do concur with my observation, does it imply that Zfh2 is translocated in the nucleus in after stress? Again, this is just my curiosity.

4. It would be very useful to include scale bars in all the confocal images, and include a clear description of what the arrowheads and stars indicate throughout of figure legend. The authors did so for very few panels. One can deduce what they point to, but the readers should not have to.

5. Figure 4 panels do not match the figure legend description. Everything is shifted, 4A in the figure is 4B in the legend, 4B is 4C, 4C is 4A.

6. There is no figure 6G (page 14, line 295). Do the authors mean 6B?

**Have all data underlying the figures and results presented in the manuscript been provided?**

Reviewer #1: No: I did not find an attached spreadsheet of the numerical data or statistics

Reviewer #2: No:

PLOS authors have the option to publish the peer review history of their article (what does this mean?). If published, this will include your full peer review and any attached files.

Reviewer #1: No

Reviewer #2: Yes: François Leulier

---

## [Decision Letter · Decision Letter 1]

22 Nov 2019

Dear Dr Biteau,

Thank you very much for submitting your Research Article entitled 'Zfh2 controls progenitor cell activation and differentiation in the adult Drosophila intestinal lineage' to PLOS Genetics. Your manuscript was fully evaluated at the editorial level and by independent peer reviewers. As you will see from the reviewer comments, Reviewer 2 was satisfied with this revised version, whereas Reviewer 1 points out several outstanding issues. We therefore ask you to modify the manuscript as follows, before we can consider your manuscript for acceptance:

1. remove the p4E-BP western blots (Suppl. Fig 4B) and corresponding text, because they are not convincing. It is not clear which band is the correct band (the one that goes up upon Rheb overexpression may be correct, but this is circular reasoning), the loading between lanes is not equal, total 4E-BP is missing, and overall the signal/noise ratio is poor.

2. Remove Ras/MAPK signaling from the model in Fig. 8 because you do not seem to have any data that Zfh2 affects Ras/Mapk signaling. The data simply show Ras/MAPK signaling is required for growth, but so is any gene required for cell growth or proliferation (E2F, actin, gadph, etc).

3. Separate out the Delta staining in Fig. 2A as suggested by Reviewer 1.

[LINK]

Yours sincerely,

Aurelio A Teleman

Associate Editor

PLOS Genetics

Gregory P. Copenhaver

Editor-in-Chief

PLOS Genetics

Reviewer's Responses to Questions

**Comments to the Authors:**

Reviewer #1: The authors have submitted a revised manuscript that further supports their conclusions that Zfh2 is required and is sufficient to drive EB activation, which includes cellular growth, cell shape change, transient membrane protrusion formation and increased compensatory ISC proliferation. For example, they have provided new data showing that Zfh2 is required for EB activation upon other stresses (e.g. infection, oxidative stress). They have also provided some of the requested controls (e.g. antibody validations) for their experiments and have edited their figures for more clarity. Overall, the data showing Zfh2’s role in EB activation are convincing; however, the mechanistic insight into how Zfh2 regulates EB activation/differentiation is disappointedly still weak. The claim that Zfh2 is upstream of TOR and MAPK and parallel to insulin signaling is unfortunately still not well supported.

I’ve outlined my concerns below:

1. The authors state that Zfh2 moderately activates TOR signaling but their quantification from their immunostainings and Western blots in Figs. 5A and S4A shows only a 2-fold increase. Furthermore, the increase in phospho-4EBP levels after Rheb overexpression is even less than from Zfh2 overexpression. This is surprising since Rheb is the strongest known positive regulator of TOR signaling. The increases in phospho-4EBP (barely 2-fold) after Rheb overexpression do not reflect the TOR activation shown by previous Rheb overexpression studies (e.g. Saucedo et al., 2003), which show a dramatic increase. As stated in my previous review, the variation of phospho-4EBP levels on the Western blots in Figure S4B after Rheb overexpression is concerning as Rheb overexpression should consistently increase TOR activity and thus phosphor-4EBP levels. Reviewer 2 also expressed concerns about these Western blots. Even if TOR activity was increased 2-fold by Zfh2, it’s not clear whether this amount of TOR activity can effect EB activation. Additionally, the phospho-4EBP staining in Figure S4A is not consistent with the Western blots in S4B. The phospho-4EBP levels in Fig S4A do not surprisingly increase in all cells (ISC and EB pairs) in the field after Rheb overexpression for 3 days. If there are fewer progenitors positive for TOR activation per midgut and Rheb overexpression doesn’t increase TOR signaling as much as Zfh2 overexpression, then why does the Western blot quantification (Fig. S4B) show that Rheb overexpression increases TOR activity more than Zfh2 overexpression? Also to note, the WB quantification does not seem to match what is observed in the provided Western blots. Altogether, this data does not convincingly show that Zfh2 moderately increases TOR activity and whether this activation is effective for EB activation. Lastly, as mentioned in my previous review, the requirement for TOR (an essential regulator of many cellular processes including protein translation) for Zfh2 to promote EB growth is not surprising.

2. The data suggest that InR, MAPK and TOR signaling are sufficient to promote EB growth and that insulin signaling can also promote non-autonomous ISC proliferation as published by Choi et al., 2011. The data also suggest that Ras and TOR signaling are required for Zfh2-mediated EB growth. However, the authors do not identify how Zfh2 promotes cell shape change, protrusion formation and non-autonomous ISC proliferation (which they show does not require insulin signaling). Additionally, the data suggest that insulin, Ras and Rheb signaling are not sufficient to drive full EB activation. The manuscript is about EB activation by Zfh2, which they have defined to include cellular growth, cell shape change and protrusion formation. Although Ras and Tor signaling seem to be required for Zfh2 induced EB growth, it’s not convincing from the data provided if they are required downstream or in parallel to Zfh2 activity. Convincing evidence that Zfh2 activity can activate Ras and TOR signaling and including parallel genetic experiments with controls that determine that Tor and Ras are downstream (and not parallel) to Zfh2 would strengthen the manuscript.

3. The authors suggest that E-cadherin may be affected by Zfh2 activity in EBs. Choi et al., 2011 suggest that insulin signaling in EBs downregulates E-cadherin level between the ISC and EB, thus triggering ISC proliferation. Thus, testing whether Zfh2 is required for E-caherin downregulation could support its role in non-autonomous induction of ISC proliferation, possibly through insulin signaling.

4. Their data suggests that Zfh2 is required in EBs that have received a Notch signal (GBE+) for their growth and differentiation. Whether TOR is strongly activated by Zfh2 is not clear from the provided data and the data also doesn’t provide a mechanism by which TOR is activated by Zfh2. I suggested in my previous review that the authors could check whether Zfh2 loss results in increased TSC2 levels. It has been reported that TSC2 is downregulated in EBs by a Notch signal (Kapuria et al., 2012), but how this occurs is not known.

Minor comments:

1. The Delta staining in Fig. 2A is not visible. I suggest separating the channels so that the Delta staining can be clearly shown. If the purpose for this panel is to show how each cell-type was identified for quantification, the control could be included in the main figure and the other genotypes could be placed into a supplementary figure.

2. The data supports that Zfh2 is important for absorptive lineage-progenitor activation. Thus, the title of the manuscript should include the words “absorptive lineage progenitor cell activation.” I would also discuss that EE-progenitors may also require activation.

Reviewer #2: The authors answer to my previous comments in a satisfactory way and as a result the manuscript has improved. I feel the story is now of sufficient interest and quality to be published.

**Have all data underlying the figures and results presented in the manuscript been provided?**

Reviewer #1: Yes

Reviewer #2: Yes

PLOS authors have the option to publish the peer review history of their article (what does this mean?). If published, this will include your full peer review and any attached files.

Reviewer #1: No

Reviewer #2: Yes: François Leulier

---

## [Editor Report · Decision Letter 2]

5 Dec 2019

Dear Dr Biteau,

We are pleased to inform you that your manuscript entitled "Zfh2 controls progenitor cell activation and differentiation in the adult Drosophila intestinal absorptive lineage" has been editorially accepted for publication in PLOS Genetics. Congratulations!

Yours sincerely,

Aurelio A Teleman

Associate Editor

PLOS Genetics

Gregory P. Copenhaver

Editor-in-Chief

PLOS Genetics

Comments from the reviewers (if applicable):

**Data Deposition**

http://datadryad.org/submit?journalID=pgenetics&manu=PGENETICS-D-19-00930R2

Press Queries

---

## [Editor Report · Acceptance letter]

10 Dec 2019

PGENETICS-D-19-00930R2 

Zfh2 controls progenitor cell activation and differentiation in the adult *Drosophila* intestinal absorptive lineage 

Dear Dr Biteau, 

We are pleased to inform you that your manuscript entitled "Zfh2 controls progenitor cell activation and differentiation in the adult *Drosophila* intestinal absorptive lineage" has been formally accepted for publication in PLOS Genetics! Your manuscript is now with our production department and you will be notified of the publication date in due course.

With kind regards,

Nicholas White

PLOS Genetics

On behalf of:
